



# Volcanism straddling the Mio-Pliocene boundary on Patmos (East Aegean Sea): Insights from new $^{40}$Ar/$^{39}$Ar ages

**Katharina Boehm[1], Klaudia F. Kuiper[1], Bora Uzel[2], Pieter Z. Vroon[1], and Jan R. Wijbrans[1]**

[1] Vrije Universiteit Amsterdam, Department of Earth Sciences, 1081HV Amsterdam, the Netherlands

5   [2] Dokuz Eylül University, Department of Geological Engineering, TR-35160 İzmir, Turkey

*Corresponding to: Klaudia Kuiper(k.f.kuiper@vu.nll)*





**Abstract.** The island of Patmos, in the eastern Aegean Sea, consists almost entirely of late Miocene to Pliocene volcanic rocks. The magmatism in the Aegean is governed by subduction of the African plate below the Eurasian plate, back-arc extension, slab roll-back, slab edge processes and westward extrusion of central Anatolia to the west along the Northern Anatolian Fault into the Aegean domain, The evolution of the Aegean basin is that of a back arc setting, with a southerly trend in the locus of both convergent tectonics, and back arc stretching, allowing intermittent upwelling of arc, lithospheric and asthenospheric magmas.

Here, we present new $^{40}Ar/^{39}Ar$ age data for Patmos and the nearby small island of Chilomodi to place this volcanism in a new high resolution geochronological framework. High resolution geochronology provides a key to understanding the mechanisms of both the tectonic and magmatic processes that cause the extrusion of magma locally, and sheds light on the tectonic evolution of the larger region of the back-arc basin as a whole.

The volcanic series on Patmos is alkalic, consistent with a back arc extensional setting and ranges from trachybasalt, to phonolites, trachytes and rhyolites, with $SiO_2$ ranging from 51.6 -80.5 wt.% and $K_2O$ from 2 -11.8 wt.% with extrusion ages ranging from 6.59±0.14 Ma – 5.17 ± 0.11 Ma. Volcanism on Patmos and adjacent Chilomodi can be understood by a combination of mantle and crustal tectonic processes including influence of transform faults and rotational crustal forces that also caused the opening of the south Aegean basin due to roll back of the subducting slab south of Crete.

## 1 Introduction

In the Aegean and Western Anatolia region Cenozoic magmatism is widespread and compositionally diverse. The evolution of the numerous volcanic centres is governed by subduction-related, back-arc and intra-plate magmatic processes. The region has been subjected to compressional and extensional forces governed by the subduction of continental platform sediments and roll-back of the subducting slab, that caused a gradual shift of the trench and trench related processes to the south, forming the Aegean as an extensional, back-arc basin (McKenzie, 1978; Le Pichon and Angelier, 1981; Horvath and Berckhemer, 1982). Typically, in a back arc basin the overall stress field is extensional, allowing upwelling of magmas originating from deep sources. At the same time in back arc settings the relics of products of earlier compressional processes survive at depth adding to diversification and mingling of the magmatic signatures of volcanic products. Understanding of such volcanism adds to improved insights into the tectonic processes that underly the formation of the Aegean back-arc basin, one of the best studied back-arc basins globally (Jolivet and Brun, 2010; Van Hinsbergen and Schmid, 2012; Ring et al., 2017).

The island of Patmos (~34 km² in surface area) is one of several volcanic centres in the Aegean. Patmos is part of the Dodecanese island group, at present situated 100 km NW of Nisyros which represents the present-day locus of arc volcanism in the Eastern Aegean Sea (Fig. 1 (a)). Located centrally between the two largest metamorphic core complexes in the region, that of the Cyclades and the Menderes, which both experienced extreme crustal thinning by low angle normal faulting during the middle Miocene (Bozkurt and Mittwede, 2005; Van Hinsbergen and Schmid, 2012; Rochet et al., 2018), Patmos was located on thinned continental and back-arc crust and lithosphere by the end of the Miocene. Immediately to the south of



Patmos around the Mio-Pliocene boundary, the opening of the south Aegean basin started forcing Crete further to the south, and causing rotation with Euler poles just east of the Peloponnese in the west and just south of Patmos off the Anatolian coast line in the east (Kissel and Laj, 1988; Duermeijer et al., 1998). By the end of the Miocene the adjacent Cycladic and Menderes core complexes switched from ductile lower block accommodated low-angle normal faulting to steepened brittle normal faulting accommodated horst graben tectonics. Volcanism on Patmos was active during the Messinian (late Miocene)

and the early Zanclean (Pliocene), a period of anomalously low water tables throughout the Mediterranean domain which cause decompression in the already thinned crust underlying Patmos Island.

The island of Patmos is almost entirely volcanic and offers a range of potassium-rich rock types, including trachyandesites, trachytes, phonolites, and rhyolites (Robert, 1973; Fig. 1 (b)). The oldest volcanic rocks on Patmos are phonolites; with other volcanic rocks roughly 1 Ma younger (Wyers, 1987).


In this paper we provide new high resolution $^{40}$Ar/$^{39}$Ar ages for the different volcanic units on Patmos and the small neighbouring island of Chilomodi. These new data allow us to complement and refine existing datasets from Wyers (1987), as well as to place these results in a tectonic context where we will assess the possible roles of (a) slab roll-back and contemporaneous back-arc extension (e.g. Palmer et al., 2019; Pe-Piper and Piper, 2007), (b) possible periodic upwelling of

sub-continental lithospheric or asthenospheric melts through gap(s) in the subducting slab, and (c) tectonic regime changes in the area at the time. The westward propagation of the Northern Anatolian Fault (NAF) into the Aegean around 5 Ma resulted in changes of extensional tectonics and fault patterns (Armijo et al., 1999; Armijo et al., 2002). Here we want to constrain the potential links between changing tectonic stress fields and volcanism on Patmos.

## 2 Methodology


Samples were collected from different units both on Patmos and Chilomodi, based on the Geological map of Patmos (Galeos, 1993), the thesis of Wyers (1987) and the publications Wyers and Barton (1986, 1987) (see coordinates in Table 1). All sample preparations and analyses of this study were performed at Vrije Universiteit Amsterdam, the Netherlands. Unaltered, cleaned samples were crushed and subsequently split into fractions for mineral separation for $^{40}$Ar/$^{39}$Ar

geochronology, and for powdering for major elements. We analyzed each sample for major elements and measured $^{40}$Ar/$^{39}$Ar ages on sanidine and/or biotite phenocrysts.

### 2.1 Major elements

Major element concentrations were measured by X-ray fluorescence spectroscopy (XRF) on fused glass beads on a Panalytical AxiosMax instrument. Sample powders were ignited at 1000°C for 2 hours to determine the loss on ignition

(LOI), before being mixed with $Li_2B_4O7$/$LiBO_2$ (1:4 dilution) and fused to glass beads at 1150°C. All XRF results are



reported on a volatile-free basis and normalized to 100 wt.% and Fe is expressed as total ferrous iron (FeO*). Precision and accuracy for major elements were better than 2% RSD.

## 2.2 $^{40}$Ar/$^{39}$Ar Geochronology

Sanidine and/or biotite separates were obtained using standard heave mineral separation procedures (Frantz magnet, heavy liquid separation). A table-top Jeol SEM was used to assess potassium contents of the minerals. Samples were hand-picked under an optical microscope, leached with diluted $HNO_3$ and cleaned with deionized water in an ultrasonic bath. Samples and standards (Fish Canyon sanidine) were wrapped in Al –foil, loaded in Al-cups and irradiated for 12 hours in the CLICIT facility of the OSU TRIGA reactor at Oregon State University (VU114). After irradiation multiple grains samples and

standards were loaded into a copper disk with holes of 2mm diameter, pre-heated under vacuum for >24 hours at 250°C and heated overnight at 120° in an ultra-high vacuum extraction line. The mineral grains were fused at 12% beam intensity of a focused 50W continuous wave Synrad 48-5 series $CO_2$ laser system. The released gas was cleaned using hot getters (SAES NP10, ST172, Ti sponge) and a Lauda cold trap at -70°C to eliminate reactive gases. The purified Ar was analysed isotopically on a ThermoFisher Helix MC multi-collector noble gas mass spectrometer, with two Faraday collectors with

$10^{13}$ Ohm resistor amplifiers ($^{40}$Ar and $^{39}$Ar beams) and three pulse counting CDD collectors ($^{38}$Ar, $^{37}$Ar and $^{36}$Ar beams). The H2, H1, AX and L1 detectors (m/e: 40, 39, 38 and 37) are fitted with standard slits resulting in a resolution of ca 850 to resolve hydrocarbon interferences from the argon isotope signals and the L2 channel (m/e: 36) was fitted with a high resolution slit (resolution >1500 to resolve $^{12}C_3$ and $^1H^{35}Cl$ from $^{36}Ar$). Gain calibration for the Faraday and CDD cups was done by peak jumping of an air $^{40}$Ar beam (~50fA) on all detectors in static mode. Intensities are regressed to time zero and

gain calibration factors are determined relative to the AX-CDD. Between the sample measurements system blanks were measured regularly every 3 steps and air pipette aliquots of $^{40}$Ar/$^{36}$Ar were analysed for monitoring the mass discrimination. The software ArArCALC2.5 (Koppers, 2002) and the air ratio of 298.56 ± 0.31 of Lee et al. (2006) were used. Interfering isotope production ratios can be found in Table A1. The reported full external error includes analytical uncertainty and includes error in J-values, standard age and decay constants with 2 sigma uncertainty (2σ).


## 3 Results

### 3.1 Petrography

The trachybasalt (P2) sampled on Chilomodi contains pristine olivine (ol), clinopyroxene (cpx), plagioclase (plag) and

oxides. It also contains gabbroitic microcumulates of plag, ol and cpx. The sample has a coarse matrix, and does not show any signs of alteration (no sericite in plag, no iddingsite in ol).



The basaltic trachyandesite (P17) contains about 35% plag, 30% K-feldspar, 20% sodalite, 10% nepheline, 5% leucite. It has a coarse plag rich groundmass (grm) and a porphyritic texture.

The most common phenocrysts observed in the trachytes of Patmos and Chilomodi are K-feldspar (kfsp), pyroxene (px), biotite (bt), plagioclase (plag) in a porphyritic groundmass. Some trachytes have big kfsp phenocrystals (e.g. P4, P7). The trachyte (P11) has a fine qtz-bt matrix with phenocrysts of plag (~40%), quartz (qtz) (~30%), bt (~20%) and kfsp (~10%). Trachyte P13 has similar mineralogy, but shows subhedral plag crystals. The trachyte P7 does not show signs of alteration and contains ~30% quartz (qtz) and ~20% kfsp as well as cpx, bt and ol. The porphyric texture is dominated by big phenocrysts of kfsp and cpx and ~35% grm. It also contains cumulates of ol, px, and fsp. Olivine (15%) is also observed in the trachyte P16.

The rhyolites have relatively few crystals, mostly kfsp and cpx phenocrysts. Rhyolite P1 also contains plag and bt, in rhyolite P3 also zeolites are observed, rhyolite P5 is very altered and contains mainly clay minerals and rhyolite P6 is also a slightly altered. P8 is a glassy rhyolite which also contains titanite and an altered groundmass (clay minerals). The rhyolite P9 is a relatively fresh rock, is very fine grained and contains ~49% glass, ~35% kfsp, 15% bt and ~1% cpx.

The phonolites sampled on Patmos contain phenocrysts and clusters of kfsp (~25-39%), sodalite (~15-35%), chlorite (~20-30%), amphibole (8-10%), biotite (up to 15% in P10B), nepheline (P12), orthopyroxene, clinopyroxene and titanite (P10A, P12). The matrix consists mainly of plag and chlorite and is much finer and contains more glass in P10B than P10A. Thin section photos can be found in the supplement (Fig. S1).

## 3.2 Major element chemistry

Results of major element XRF analyses are listed in Table 3. The $SiO_2$ content in the selected volcanic rock samples ranges from 51.6 to 80.5 wt.%. The total alkali content ($K_2O+Na_2O$) of the samples ranges from 6.4 to 14.5 wt.%, $K_2O$ varies between 2 wt% and 11.8 wt.% and $Na_2O$ between 1.4 to 7.9 wt.% (except P15 which has 0.11 wt.% $Na_2O$). Accordingly, eight samples classify as rhyolites (P1B, P3, P5, P6 from Chilomodi and P6, P8, P9, P15 from Patmos), five as trachytes (P4 from Chilomodi and P7, P13, P16, P19, P11 from Patmos), three as phonolites (P10A, P10B, P12 from Northern Patmos), one as andesite (P1A enclave in P1B), one as basaltic trachyandesite (P17 dike in Southern Patmos) and one as trachybasalt (P2 from Chilomodi), (TAS diagram Fig. 2 (a); Le Bas et al., 1986). The $K_2O/Na_2O$ ranges from 0.45 to 8.35 (P15 72.61). The trachytes P11, P16, P19 and the rhyolites P8 and P9 have the lowest $Na_2O$ content and thus the highest $K_2O/Na_2O$ ratios, while the phonolites are the most sodic rocks on Patmos ($K_2O/Na_2O<1$) (Fig. 2 (b)).





### 3.3 Argon geochronology

The $^{40}$Ar/$^{39}$Ar results indicate that active volcanism on Patmos occurred between 5.7 Ma and 6.0 Ma with a first pulse of activity around 6.5 Ma (Fig. 3; supplementary information Table S1). Nearby Chilomodi island was active from 5.2 Ma to 5.4 Ma. Rhyolite P1 yields a total fusion age of 5.17 ± 0.11 Ma (full external error, 2σ). The radiogenic $^{40}$Ar$^*$ was on average
87% (77-94%) and the MSWD 0.46 (n=19/20). Rhyolite P3 had rather low K contents for sanidine with values ranging between ~2.0-2.6 wt.% K (based on pre-irradiation sample screening on table-top SEM) with K/Ca ratios of ~14-15 (based on conversion of the measured $^{39}$Ar$_K$/$^{37}$Ar$_{Ca}$ ratio). Two size fractions yielded relatively low radiogenic $^{40}$Ar$^*$ yields of ~35-49%. The smaller size fraction of 355-500 μm had slightly lower radiogenic $^{40}$Ar$^*$ (~35%) and resulted in a total fusion age of 5.34±0.12 Ma, while the coarser fraction of 500-1000 μm resulted in 5.40±0.12 Ma (MSWD 0.18 and 0.27, n=18/19 and
n=18/20, respectively). Two size fractions from rhyolite P4 yielded 5.39±0.13 Ma (355-500 μm) and 5.41±0.12 Ma (500-1000 μm). The radiogenic $^{40}$Ar$^*$ was on average 31% and 59% for respectively the finer and coarser fraction (MSWD 0.04 and 0.44, n=13/15 and 14/20). The rhyolite P6 yielded 5.33±0.11 Ma ($^{40}$Ar$^*$ ~62%, MSWD 0.27, n=14/15).

The rhyolite and trachytes of Patmos Island yielded slightly older ages than the rhyolites from Chilomodi. Rhyolite P9
resulted in 5.98±0.13 Ma ($^{40}$Ar$^*$ ~72%, MSWD 1.71, n=12/15). Trachyte P7 resulted in a sanidine age of 5.63±0.12 Ma ($^{40}$Ar$^*$ 80%, MSWD 0.37, n=20/20) and a biotite age of 5.84 ± 0.13 Ma ($^{40}$Ar$^*$ 71%, MSWD 0.51, n=10/15). Three separates of trachyte P11 were measured. The diamagnetic sanidine fraction yielded 5.69±0.12 Ma and the slightly more magnetic sanidine yielded 5.84±0.12 Ma. The second sanidine fraction had a higher $^{40}$Ar$^*$ yield (88% average) than the first sanidine fraction (80% average). The biotites appeared to be too altered and did not yield reliable ages. Their $^{40}$Ar$^*$ yield was only
13% indicating severe alteration and therefore, the biotite data are excluded. Trachyte P13 yielded a weighted mean age for biotite of 5.87±0.13 Ma ($^{40}$Ar$^*$ 60%, MSWD of 1.23, n=13/15), which due to its high $^{40}$Ar$^*$ content was considered to be the only analytically reliable biotite age result.

Phonolites are the oldest rocks of this study. Two fractions of 355-500 μm sanidine were dated from sample P10. The first
fraction contained the non-magnetic sanidines, while the second fraction contains slightly more magnetic sanidines of the same grain size range. The most non-magnetic and slightly magnetic fraction yielded total fusion ages of 6.54±0.15 Ma ($^{40}$Ar$^*$ 64%, MSWD of 2.73, n=19) and 6.59±0.14 Ma ($^{40}$Ar$^*$ 96%, MSWD 0.38, n=15). Note, that none of the samples in this studie shows evidence of excess argon.



## 4 Discussion

### 4.1 Reproducibility of $^{40}$Ar/$^{39}$Ar ages

We analysed both sanidine and biotite of Patmos sample P7 yielding a biotite age that is ~0.21Ma older than the sanidine. Coevally erupted sanidines and biotites indeed do not always match in age (e.g. Kuiper et al., 2004; Hora et al., 2010). This can be explained by (1) Recoil from K-depleted cleavage zones in partially altered biotites (Smith et al., 2008). Alteration causes the transition of K-bearing biotite laminae to K-free chlorite laminae which are more open to recoil loss of $^{39}$Ar, and thus with the leakage of $^{39}$Ar from the crystal, the apparent $^{40}$Ar/$^{39}$Ar ratio increases leading to higher observed $^{40}$Ar/$^{39}$Ar ages. Typically such processes are accompanied by the uptake of substantial amounts of atmospheric argon; (2) partial closure of biotite before the eruption, while sanidine remains open due to its lower closure temperature and can consequently degas efficiently for longer in the magma chamber (Hora et al., 2010).This process depends on if (partial) isotopic equilibrium with the atmosphere was reached at modest pressure (shallow pre-eruption storage) and it depends on cooling rates at the time of the eruption. If the closure temperature of biotite is higher than the eruption temperature, excess $^{40}$Ar can be locked in the biotite and in-situ production of $^{40}$Ar* starts, i.e. the clock starts before the eruption. Extraneous $^{40}$Ar in biotite can thus lead to older ages than in sanidine, i.e. pre-eruption ages. The ability to analyze this difference between the biotite and sanidine system proves the low limit of analytical uncertainty of the measurements.

We also analysed multiple sanidine sample splits with either different grain sizes or different magnetic behaviour. We did not observe a relation between grain size and age. However, differences in magnetic properties seem to yield different ages with the least magnetic (or diamagnetic) sanidine yielding younger ages with slightly lower $^{40}$Ar$^*$ contents (Fig. 3). Pure sanidine is weakly diamagnetic, but small amounts of Fe cations or magnetic grains/inclusions can result in increased magnetic susceptibility (e.g. Biedermann et al., 2016). Paramagnetic impurity ions can be incorporated into the feldspar structure, e.g. $Fe^{3+}$ can replace $Al^{3+}$. Magnetite may form by exsolution of Fe from the feldspar structure, usually described as response to cooling from high temperatures (Biedermann et al., 2016; Hounslow and Morton, 2004 and references therein). Small amounts of submicroscopic ferromagnetic mineral inclusions e.g. magnetite are described for sanidine, often occurring along grain boundaries and cleavage domains (Finch and Klein, 1999). From this, we infer that the sanidine grains with slightly higher magnetic susceptibility are prone to contain sub-microscopic ferromagnetic inclusions and these, we hypothesize, may potentially contain sub-microscopic inclusions which may either contain excess Ar, or play a role in hosting recoiled $^{39}$Ar produced by neutron activation. Anderson et al. (2000) described magnetite inclusions in sanidine which might have equilibrated via melt channels that formed along cleavage planes. If such an equilibration occurred minor amounts of ambient $^{40}$Ar may have entered the crystal via melt channels.

In addition, we observed variation in radiogenic $^{40}$Ar$^*$ within and between different grainsizes from the same sample. Sanidine of P3 (355-500 μm and 500-1000 μm) from Chilomodi island yielded resp. 22-44 % and 25-93% $^{40}$Ar$^*$. As this





variation in radiogenic $^{40}Ar^*$ is not linked to any age variation (Fig. 3), we conclude that the ages obtained from this sample are reproducible. $^{40}Ar/^{36}Ar$ isochron intercepts of those samples overlap with $^{40}Ar/^{36}Ar$ atmospheric ratios, so the variability

in radiogenic $^{40}Ar^*$ was unlikely caused by excess argon (e.g. from fluid inclusions). Therefore, we infer that inclusions containing atmospheric argon were present in these mineral fractions and this higher contribution of atmospheric Ar resulted in lower $^{40}Ar^*$ percentage.

In summary, while we note some significant spread in ages, the internal reproducibility within each sample fraction is excellent.


## 4.2 Geochronological framework

Our results indicate that the volcanism on Patmos and Chilomodi occurred during a brief period of 1.4 Ma with at least three distinct intervals of activity. The oldest volcanic activity yielded phonolites at ~6.5 Ma, followed by the rhyolitic and trachytic volcanism on Patmos island around ~6.0 Ma and volcanic activity on Chilomodi island around ~5.8 Ma.

In contrast with Fytikas et al., (1976) and Wyers (1987), we have not found evidence for >6.0 Ma ages in the trachytic rocks. The first K/Ar ages for Patmos were reported by Fytikas et al. (1976). They obtained $4.38 \pm 0.15$ Ma for total rock sample of an alkali basalt lava flow of Chilomodi (PAT-12), $7.03 \pm 0.025$ Ma for biotite and groundmass of a trachyte lava dome of Lefki Bay (PAT-34) and $7.20 \pm 0.025$Ma for biotite of a trachytic intrusive facies of Prasso Mt. (PAT-28). Wyers (1987) dated five whole rock (WR) samples (rhyolite, phonolites, ne-trachybasalt, hy-trachybasalt, hy-trachyandesite) with the K/Ar

method and three WR samples (trachyte and young and older ne-trachybasalt) with the $^{40}Ar/^{39}Ar$ method. Decay constants and standard conventions were not reported and thus direct comparison is difficult, leading to an additional uncertainty between the older and the new results of >1%. Further, the whole rock approach has shown difficulties and modern $^{40}Ar/^{39}Ar$ geochronology dates either groundmass (without phenocrysts) or the K-bearing phenocrysts in volcanic rocks. Moreover, the youngest samples (4.49-4.64Ma) of Wyers (1987) have very low radiogenic $^{40}Ar^*$ yields (<32 %) and suggest alteration

and/or Ar loss similar to biotite sample P11. Due to these uncertainties, these previously published ages are not included in the present discussion.

## 4.3 Regional dispersion of phonolites

Phonolites derive from silica-undersaturated, mafic magmas and are commonly found in intra-plate settings, intra-

continental rifts and oceanic island settings. The phonolites of Patmos are a rare example of phonolites occurrence in or near a volcanic arc or back-arc setting. In Western Anatolia, west of the current slab gap as indicated by tomography (e.g. Biryol et al., 2011; Fig. 1 (a)), the volcanic centre of Foca, is one of the few nearby localities where phonolites are described (Akay and Erdoğan, 2004; Altunkaynak et al., 2010). Altunkaynak et al. (2010) performed laser step heating of bulk plagioclase of a phonolite (SF-11) and reports an age of 14.12 Ma, significantly older than the ones on Patmos. Besides the phonolites of





Foca, in proximity of Isparta, below the tomographically indicated slab gap, two occurrences of phonolites are described in the literature. In the area of Afyon (Akal et al., 2013; Prelević et al., 2015) and Senirkent graben (Elitok, 2019) phonolites were studied but no absolute high-precision time constraints were published so far. In summary, although a minor component, several occurrences of Miocene phonolites are described in the eastern Aegean and western Anatolian volcanic province.


## 4.4 Alkalinity of the volcanism

The ratio of $K_2O/Na_2O$ demonstrates the chronological evolution from sodic for the oldest rocks of Patmos (phonolites), to potassic (trachytes Patmos island), and again to (almost) sodic nature (Chilomodi rhyolites; Fig. 4). The sodic character of the phonolites is one characteristic of a possible asthenospheric influence. Although high $K_2O+Na_2O$ volcanics are the norm,

the Chilomodi rhyolite (P1) also contains andesitic enclaves that represent quenched magma compositions with a lower $K_2O+Na_2O$ content, comparable to the trachybasalt of Chilomodi and the volcanics from Nisyros and Kolumbo (Fig. 2 (b)). The volcanic rocks of Chilomodi island have similar $Na_2O$ contents as the volcanics of SAVA and Kula (Fig. 2 (b)). The trend in the youngest volcanics of our study (Chillomodi trachybasalt, trachyte, rhyolites and enclave) is in agreement with the general more sodic trend observed towards the Quaternary, which is also manifested in the relatively sodic nature of the

Kula basalts.

## 4.5 Tectonic constraints on Late Miocene to early Pliocene volcanism in the northeastern Dodecanese, as found for Patmos

The Neogene tectonics of the Aegean realm are dominated by subduction, back-arc extension, slab roll-back, slab fracturing,

westward escape of Anatolia along the North Anatolian Fault Zone (NAFZ) and the presence of a gap between Hellenic and Cyprus slabs, hypothesized to provide a channel to deeper asthenospheric magma sources.

Patmos has tectonic features associated with extensional graben structures (Dilek and Altunkaynak, 2007). According to Roche et al. (2019), the marble basement of Patmos likely belongs to the Lower Cycladic Blueschist Nappe (LCBN). The

latter is formed in a comparable tectonic setting as the Amorgos unit and the "Menderes cover sequence" (situated in the SE and the NW) and the LCBN is overlain by the Ören unit and the equivalent Afyon zone (e.g. Van Hinsbergen and Schmid, 2012). To the north, the LCBN is overthrust (Trans Cycladic thrust) by the Upper Cycladic Blueschist Nappe (UCBN) and the Dilek Nappe which is an equivalent position as the UCBN. Altogether, Patmos is situated in-between two large Metamorphic Core Complexes (MCC): the Cycladic MCC and the Menderes MCC (Fig. 1 (a)).






After early to middle Miocene core complex exhumation facilitated by large scale low angle normal faults, several authors indicate that the classic-style back-arc extension ended and was followed by a phase of N-S crustal shortening in the late Serravallian to early Pliocene (e.g. Bozkurt and Mittwede, 2005). This period of compression is explained (a) by local block rotations and (b) by the propagation of the NAFZ and the westward extrusion of Anatolia (Roche et al., 2019). In the

Messinian (Westaway et al., 2005) or Pliocene (Bozkurt and Mittwede, 2005) a new style of ~N-S extension with normal faults resulting in horst graben structures commenced and resulted in crustal thinning in the south central Aegean accommodated by rotation with Euler poles east of the Peloponnesus in the west and south east of Patmos just off the Anatolian coast (Kissel and Laj, 1988, Duermeijer et al. 1998). Tectonics accommodating the eastern rotational pole may have facilitated the ascent of magma in this area around the Mio-Pliocene boundary.


In the Dodecanese, tectonics features of the islands north and south of Patmos (e.g. Leros by Roche et al. (2018) or Samos by Ring et al. (1999)) indicate ductile deformation and nappe emplacement, followed by a short period of E-W crustal contraction, and extension related brittle E-W striking normal faulting. Tectonic studies on Patmos, however are limited. Wyers (1987) only mentions a NW-SE trending horst-graben structure and block faulting. Since horizons for correlating

volcanic eruptions on Patmos are absent, it is difficult to come to a division of units on formal stratigraphical grounds. Without information on unconformities and deformation, only limited tectonic reconstructions are possible. However, with these limitations in mind, we will attempt to sketch the tectonic evolution of Patmos in the light of our newly obtained ages, while taking into account information from regional stress fields as deduced from neighbouring islands and areas.

Stratigraphy on Patmos started with marble basement rocks and deposition of epiclastics (Galeos, 1993). On the Diakofti

Cape in the south of Patmos epiclastics occur as almost vertical banks intercalated with NW-SE trending rhyo-trachytic dykes and basic dykes. If there is no unconformity at the base of the epiclastics and if we assume that the other occurrences of epiclastics on Patmos are not tilted (Galeos, 1993), then the vertical beds within the epiclastics may be the effect of inversion in strike-slip faults (possibly related to the IBTZ, of Uzel et al., 2013).

Tectonic setting and magmatic activity are often argued to be closely related. Large scale and regional plate tectonic settings

govern the formation of magmas and local tectonic stress fields dictate whether magmas can reach the surface. The two localities where phonolites are found on Patmos lie on a NW-SE oriented strike-slip fault (for fault data see Ring et al., 2017; Fig. 1 (b) dashed lines), i.e. a conjugate-fault to the NE-SW oriented large scale transfer faults (IBTZ and Mid Cycladic Lineament). If assuming that the IBTZ is marking the western end of the slap gap between the Aegean and Cyprus slab, a crustal-scale, conjugate fault on Patmos could be an expression of the slab edge and local asthenospheric upwelling

could be easily explained.

The volcanic activity continued between 6.0 Ma and 5.6 Ma with the eruption of rhyolites and trachytes on Patmos. The localities are situated close to NW-SE oriented faults. Although crustal-scale faults are probably not changing within short time spans (of 0.5 My), it is possible that a switch from extensional to compressional tectonics (e.g. Ring et al., 1999)



blocked the path of asthenospheric upwelling. We hypothesize that high angle normal faulting became dominant in the horst-
graben development phase of post- Late Miocene extension.

At 5.4 Ma we suggest another tectonic change to E-W trending (normal) faults, which facilitated the eruption of the Chilomodi volcanics including the young basalts. Interestingly, a transition from submarine to terrestrial extrusion is described in the literature for the basalts (Galeos, 1993), which could either mean a short period of contraction (Kocygit et al., 1999) and uplift within the main stress field of extension or this transition describes a sea-level drop.


## 5 Conclusion

In this study we presented revised tectonic interpretation based on new high resolution geochronology for Patmos. We provide 12 new sanidine and 2 new biotite $^{40}$Ar/$^{39}$Ar ages on 9 different samples. With this approach we analyse biotite and
sanidine on the same sample and multiple sanidine sample splits with either different grain sizes or different magnetic behaviour. This allows us to present high quality results with high internal reproducibility within each sample fraction and this supports reliability of our age results for resolving the characteristics of the <1.5My interval of volcanism on Patmos.

Our new age data for Patmos and the nearby small island of Chilomodi indicates a brief volcanic period whith three distinct volcanic intervals. Summarizing the results, we draw the following conclusions: (1) Magmatism with an asthenospheric to
intraplate signature on Patmos started at 6.5 Ma with the eruption of sodic phonolites. A crustal-scale NW-SE oriented strike-slip fault in combination with a gap between subduction slabs below facilitated the rise of asthenospheric mantle. (2) The next phase of magmatism from 6.0 Ma to 5.6 Ma produced rhyolites and trachytes that have a more potassic nature. This could mean that the influx of asthenosphere mantle diminished and influence of sub-continental lithospheric mantle increased. This can be potentially linked to the transition from strike-slip to normal faulting resulting in a more
compressional regime. (3) Along with another tectonic change to E-W orientated faults, trachytic and rhyolitic volcanism on Chilomodi commenced at 5.4 Ma and lasted until 5.2 Ma.

## Supplement

Fig. S1 Thin section photographs
Table S1. Argon isotopic results. Corrected for background, baseline, gain, mass discrimination, reactor interferences and radioactive decay.
Table S2. Age data literature
Table S3. Compilation of literature data of phonolites and lamproites (and similar rocks) in close geographical proximity to Patmos Island.




## Author contribution

KMB, JRW and KFK were involved in the main conceptualization. KMB carried out the geochronological experiments and data reduction, with support from KFK and JRW. PZV did XRF experiments and BU provided tectonic input. KMB prepared the manuscript and visualization with reviews and editing from JRW; KMB and BU. Acquisition of the financial

support for the project was done by JRW, PZV and KFK.

*The authors declare that they have no conflict of interest.*

## Acknowledgments

Permission for fieldwork and sampling was kindly provided by the Greek Institute of Geology and Mineral Exploration.

Athanasios Godelitsas and Gregor Hofer are thanked for great assistance in the field and Roel van Elsas and Marjolein Daeter are greatly acknowledged for analytical assistance. Further, Naomi Lamers is thanked for the XRF analysis and thin section photographs. We acknowledge that this project has been funded by ISES/NWO investment grant 834.09.004

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




**Figures**



**Fig. 1 (a)** Map of the Aegean and Anatolian region modified from Biryol et al. (2011), Ersoy and Palmer (2013), Uzel et al.
(2015), Govers and Fichter (2016) and Roche et al. (2019). Recent positions of the north dipping slabs are indicated by
dashed blue iso-lines (50km distance). The blue band indicates the position of the South Aegean Active Volcanic Arc at
about 100km slab-depth (same depth is marked on the western-Cyprus slab). Between the Aegean slab and the Western
Cyprus slab, a gap, with complex vertical geometry (purple and grey dashed lines), has been observed in tomographic and
seismic data (Biryol et al., 2011; Govers and Fichter, 2016). Volcanic fields in Western Anatolia (grey areas) are compared
to Patmos volcanics in this paper. Patmos is highlighted with a green rectangle. LCB – Lower Cycladic Blueshist Unit; UCB
– Upper Cycladic Blueshist Unit; CCC – Cycladic Core Complex; MCC – Menders Core Complex, NAFZ – North
Anatolian fault zone. **(b)** Simplified geologic map of Patmos modified from Galeos et al. (1993). Our sampling sites are



indicated with symbols: Star – rhyolite, down facing triangle – trachybasalt, circle – trachyte, diamond – phonolites, square – marble. Faults are indicated as grey dashed lines.


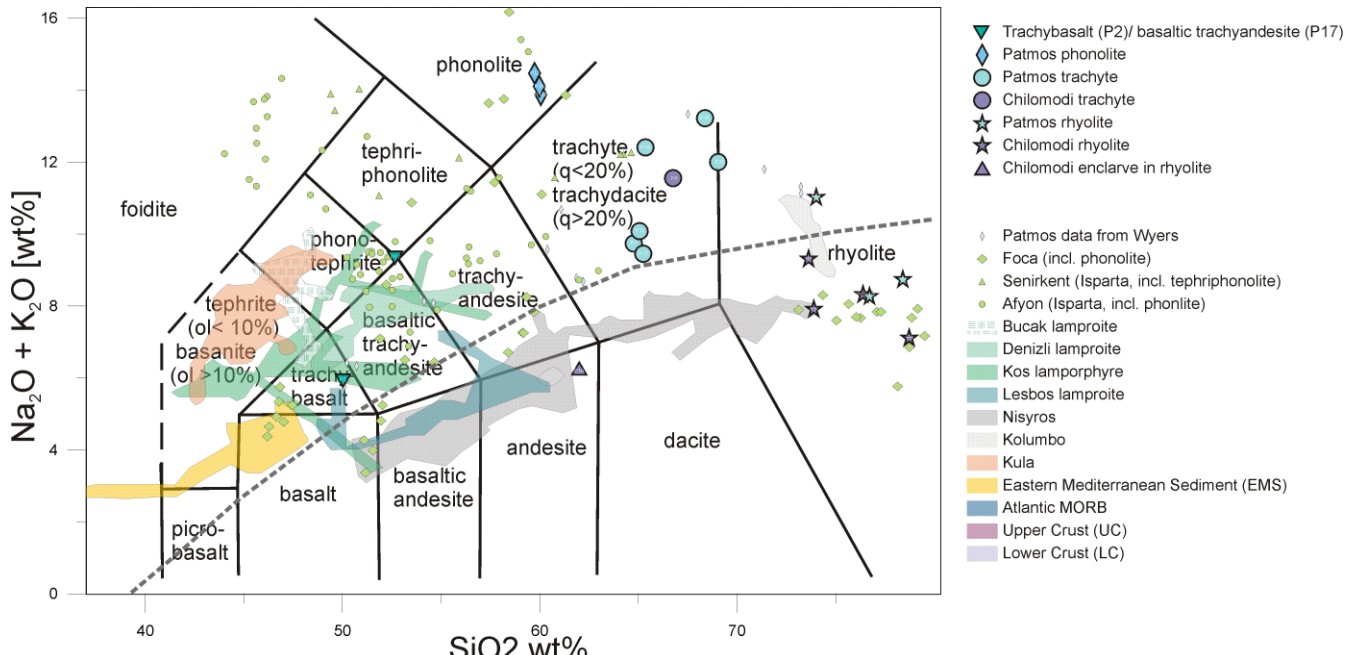

**Fig. 2 (a)** Total Alkali versus Silica diagram (TAS, after Le Bas et al., 1986) for volcanic rocks of Patmos, Eastern Aegean and Western Anatolia. Same symbols as used in Fig 1 (b). Data for fields of Western Anatolia are from Altunkaynak et al., 2010, Akay and Erdogan, 2004 (Foça); Elitok, 2019 (Senirkent); Akal et al., 2013, Prelevic et al., 2015 (Afyon); Coban and
Flower, 2007 (Bucak); Prelevic et al., 2012 (Denizli), Soder et al., 2016 (Kos); Pe-Piper et al., 2014 (Lesbos), Klaver et al., 2015, 2016, 2018 (Eastern Mediterranean Sediment EMS, Upper continental crust UC, Lower continental crust LC, Kolumbo, Nisyros); Alici et al., 2002 (Kula); Stracke et al., 2003 and references therein (Atlantic MORB).


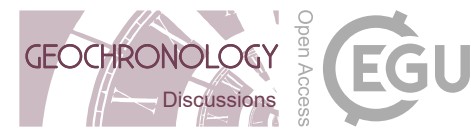

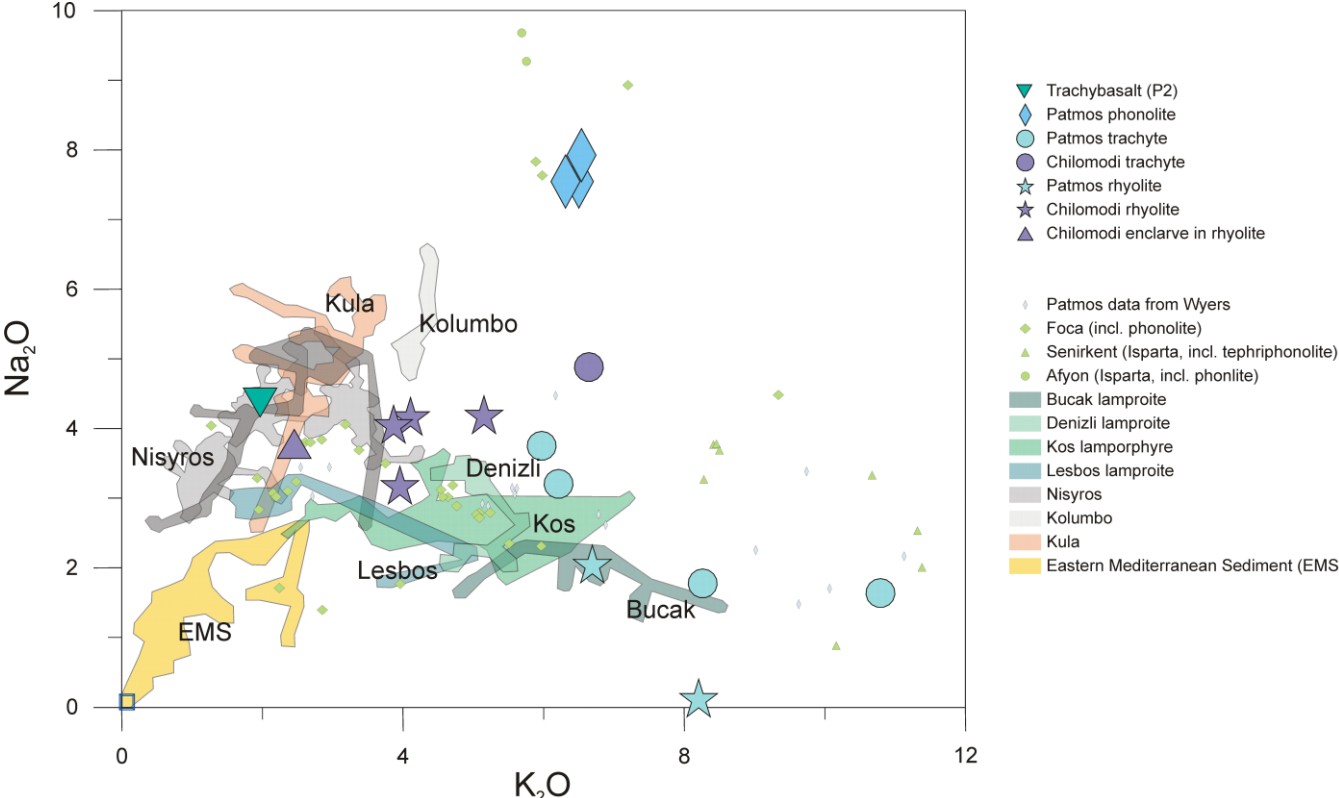

**Fig. 2 (b)** $Na_2O$ versus $K_2O$ diagram. Symbols and fields as in Fig. 2 (a). Note the large diversity in both high $Na_2O$ and $K_2O$

content of the Patmos volcanic compared to Aegean and Western Anatolian volcanic and EMS.





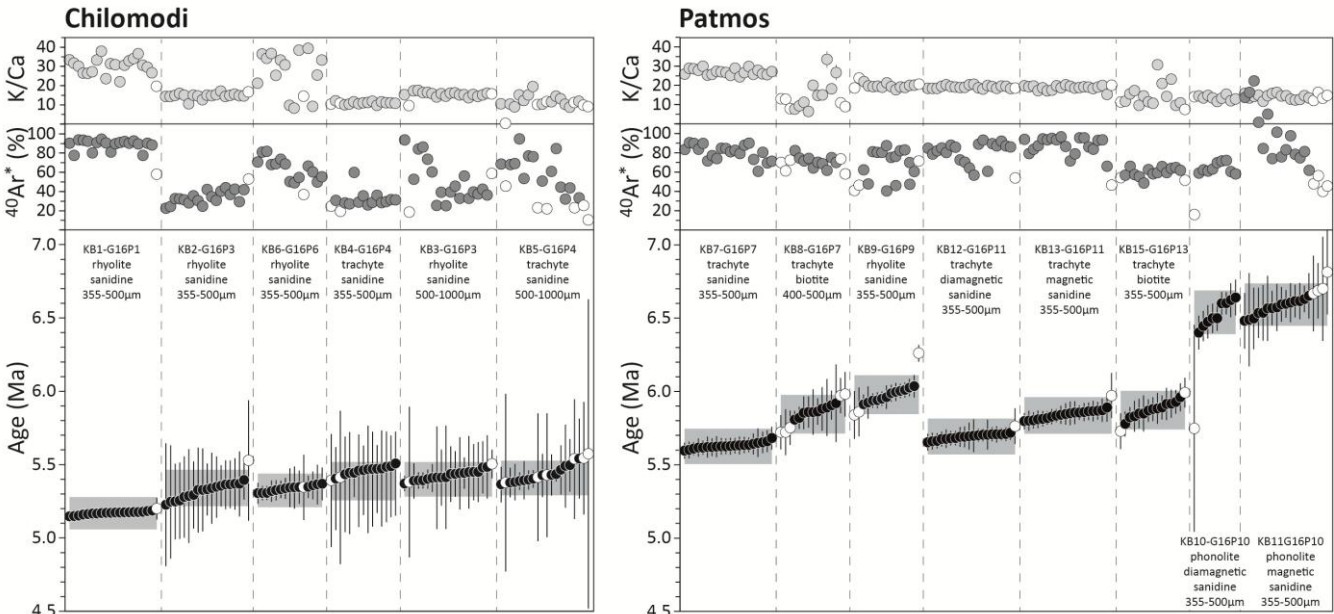

**Fig. 3** $^{40}$Ar/$^{39}$Ar diagram. Individual fusion ages with 2SD and weighted mean age.

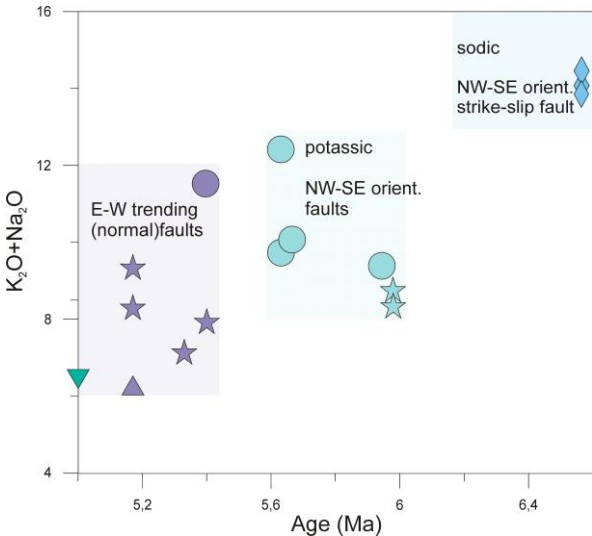


**Fig. 4** K$_2$O+Na$_2$O versus Age (in Ma). This diagram shows the volcanic rocks of Chilomodi (purple field), the volcanic rocks of Patmos (turquoise) and the phonolites of Patmos (blue). See text for discussion.



**Table captions**

**Table 1. WGS84 coordinates of samples locations on Chilomodi and Patmos**

| Sample no. | Outcrop no. | Coordinates latidude (WGS84) | Coordinates longitude (WGS84) | Discription of locality | Rock type |
|---|---|---|---|---|---|
| G16P01A | G16_P01 | 37.30805 | 26.60189 | Chillomodi, behind the church | Andesitic inclusions in rhyolite |
| G16P01B | G16_P01 | 37.30805 | 26.60189 | Chillomodi, behind the church | Glassy rhyolite, partly red oxidized |
| G16P02 | G16_P02 | 37.30757 | 26.60439 | Chillomodi, hill top | Trachybasalt, weathered platy |
| G16P03 | G16_P03 | 37.30433 | 26.60421 | Chillomodi | Rhyolite |
| G16P04 | G16_P04 | 37.30407 | 26.60456 | Chillomodi, same outcrop as P03 | Trachyte |
| G16P05 | G16_P05 | 37.30538 | 26.60384 | Chillomodi | Rhyolite, soft (clay) |
| G16P06 | G16_P06 | 37.30589 | 26.60330 | Chillomodi | Rhyolite |
| G16P07 | G16_P07 | 37.35252 | 26.53565 | N of Skala, quarry, after Military left | Trachyte very fresh |
| G16P08 | G16_P08 | 37.35146 | 26.53671 | Next to quarry of P07 | Rhyolite |
| G16P09 | G16_P09 | 37.36709 | 26.53660 | NW end of Patmos | Rhyolite |
| G16P10A | G16_P10 | 37.37302 | 26.56528 | N (middle peninsula) | Phonolite |
| G16P10B | G16_P10 | 37.37302 | 26.56528 | N (middle peninsula) | Phonolite |
| G16P11 | G16_P11 | 37.36069 | 26.55787 | N of Kampos | Trachyte |
| G16P12 | G16_P12 | 37.33780 | 26.60338 | NE peninsula, southside, ship-beach | Phonolite |
| G16P13 | G16_P13 | 37.35140 | 26.54642 | West of Kampos | Trachyte |
| G16P14 | G16_P14 | 37.29298 | 26.53527 | Marble quarry | Marble from quarry owner |
| G16P15 | G16_P15 | 37.29093 | 26.53447 | Marble quarry | Rhyolite dike in marble |
| G16P16 | G16_P16 | 37.29442 | 26.53782 | Marble quarry | Trachyte |
| G16P17 | G16_P17 | 37.28169 | 26.54934 | Souternmost Patmos | Basaltic  trachyandesite dike in rhyodacitic serie |
| G16P18 | G16_P18 | 37.28012 | 26.54773 | Souternmost Patmos | Feldspar |
| G16P19 | G16_P19 | 37.32637 | 26.53605 | NW of Skala | Trachyte |
| G16P20 | G16_P20 | 37.30053 | 26.54509 | S Patmos | (Granitic?) dike |

**Table 1.** WGS84 COORDINATES OF SAMPLE LOCATIONS ON CHILOMODI AND PATMOS. *No caption.*





## Table 2. Summary of 40Ar/39Ar results

| Parameter | G16P1 | G16P3 | G16P3 big | G16P4 | G16P4 big | G16P6 | G16P7 | G16P7 bt | G16P9 | G16P10 non-mag | G16P10 2nd | G16P11 non-mag | G16P11 2nd | G16P11 bt | G16P13 bt |
|---|---|---|---|---|---|---|---|---|---|---|---|---|---|---|---|
| Sample ID Ar | KB1 | KB2 | KB3 | KB4 | KB5 | KB6 | KB7 | KB8 | KB9 | KB10 | KB11 | KB12 | KB13 | KB14 | KB15 |
| Mineral | sanidine | sanidine | sanidine | sanidine | sanidine | sanidine | sanidine | biotite | sanidine | sanidine | sanidine | sanidine | sanidine | biotite | biotite |
| Grain size (µm) | 355-500 | 355-500 | 500-1000 | 355-500 | 500-1000 | 355-500 | 355-500 | 400-500 | 355-500 | 355-500 | 355-500 | 355-500 | 355-500 | 355-500 | 355-500 |
| Additional notes | n.a. | n.a. | n.a. | n.a. | n.a. | n.a. | n.a. | Denisty 3.05-3.22 | n.a. | Frantz max -2 | Frantz max0-max-2 | Frantz max-1.5 | Frantz max0- max -1.5 | n.a. | n.a. |
| K (wt%) (SEM) | 3.7 | 2.0 | 2.6 | 3.2 | 4.4 | 3.9 | 6.7 | 3.0 | 4.0 | 2.2 | 2.2 | 4.5 | 4.5 | 5.8 | 4.9 |
| # grains per fusion | 5 | 10 | 8 | 6 | 5 | 5 | 3 | 7 | 5 | 9 | 9 | 4 | 4 | 3 | 4 |
| Rock type | rhyolite | rhyolite | rhyolite | trachyte | trachyte | rhyolite | trachyte | trachyte | rhyolite | phonolite | phonolite | trachyte | trachyte | trachyte | trachyte |
| Locality | Chilomodi | Chilomodi P3=P4 | Chilomodi P3=P4 | Chilomodi P4 same outcrop as P3 | Chilomodi P4 same outcrop as P3 | Chilomodi | N of Skala, quarry | N of Skala, quarry | NW Patmos | N Patmos (middle) | N Patmos (middle) | N of Kampos | N of Kampos | N of Kampos | W of Kampos |
| Age (Ma) | 5.17 | 5.34 | 5.4 | 5.39 | 5.41 | 5.33 | 5.63 | 5.84 | 5.98 | 6.54 | 6.59 | 5.69 | 5.84 | 4.23 | 5.87 |
| ±2σ analytical error + J error | 0.02 | 0.05 | 0.03 | 0.07 | 0.03 | 0.03 | 0.03 | 0.05 | 0.05 | 0.06 | 0.04 | 0.03 | 0.03 | 0.13 | 0.05 |
| ±2σ full external error | 0.11 | 0.12 | 0.12 | 0.13 | 0.12 | 0.11 | 0.12 | 0.13 | 0.13 | 0.15 | 0.14 | 0.12 | 0.12 | 0.16 | 0.13 |
| MSWD | 0.46 | 0.18 | 0.27 | 0.04 | 0.44 | 0.29 | 0.37 | 0.51 | 1.71 | 2.73 | 0.38 | 0.49 | 0.73 | 0.24 | 1.23 |
| N (N total) | 19 (20) | 18 (19) | 18 (20) | 13 (15) | 14 (20) | 14 (15) | 20 (20) | 10 (15) | 12 (15) | 9 (14) | 15 (19) | 19 (20) | 19 (20) | 12 (20) | 13 (15) |
| 40Ar (%) | 88.6 | 33.5 | 50.1 | 32.4 | 61.2 | 64.1 | 80.3 | 71.3 | 73.8 | 64.2 | 98.7 | 80.7 | 87.8 | 12.7 | 60 |
| K/Ca | 29.1 | 14.2 | 15.3 | 10.8 | 11.4 | 12 | 26.8 | 8.2 | 19.6 | 13.2 | 13.8 | 19.1 | 18.5 | 54.1 | 11.6 |
| 40Ar/36Ar inverse isochrone intercept | 298.5 | 295.8 | 300.5 | 299.2 | 300.9 | 302.6 | 303.2 | 310.4 | 298.3 | 291.2 | 305.1 | 297 | 301 | 293.7 | 294.1 |
| ±2σ analytical error + J error | 6.6 | 5.4 | 2.4 | 4.2 | 3.8 | 5.1 | 6.3 | 16.4 | 6.4 | 20 | 7.8 | 5.1 | 7.1 | 14.7 | 11.5 |
| Inverse isochrone age | 5.17 | 5.43 | 5.38 | 5.37 | 5.39 | 5.29 | 5.62 | 5.77 | 5.98 | 6.62 | 6.56 | 5.7 | 5.83 | 4.7 | 5.93 |
| ±2σ analytical error + J error | 0.03 | 0.18 | 0.04 | 0.14 | 0.04 | 0.05 | 0.04 | 0.12 | 0.06 | 0.24 | 0.06 | 0.03 | 0.03 | 1.38 | 0.15 |
| ±2σ full external error | 0.11 | 0.21 | 0.12 | 0.18 | 0.12 | 0.12 | 0.12 | 0.17 | 0.14 | 0.27 | 0.15 | 0.12 | 0.13 | 1.38 | 0.2 |
| MSWD | 0.48 | 0.13 | 0.11 | 0.04 | 0.35 | 0.11 | 0.27 | 0.3 | 1.9 | 2.92 | 0.17 | 0.5 | 0.75 | 0.22 | 1.29 |
| MDF | 1.01002 | 1.01002 and 1.004184 | 1.004762 | 1.004184 | 1.004762 | 1.004184 | 1.004184 | 1.009357 | 1.009357 | 1.004184 | 1.008591 | 1.004184 | 1.004762 | 1.009357 | 1.009357 |
| 1SD (%) | 0.18 | 0.18 and 0.27 | 0.2 | 0.27 | 0.2 | 0.27 | 0.27 | 0.13 | 0.1 | 0.27 | 0.37 | 0.27 | 0.2 | 0.13 | 0.13 |
| J | 0.0017599 | 0.0017599 | 0.0018654 | 0.0017599 | 0.0018654 | 0.0017676 | 0.0017676 | 0.0018529 | 0.0018457 | 0.0017676 | 0.0018658 | 0.0017676 | 0.0018654 | 0.0018529 | 0.0018457 |
| 1SD (%) | 0.2 | 0.2 | 0.2 | 0.2 | 0.2 | 0.2 | 0.2 | 0.33 | 0.33 | 0.2 | 0.2 | 0.2 | 0.2 | 0.33 | 0.33 |


**Table 2.** SUMMARY OF [39]Ar/[40]Ar RESULTS. Includes sample characteristics, potassium content in minerals, total fusion Age, fully external error, MSWD, number of measurements, radiogenic [40]Ar, [39]Ar, K/Ca ratio, inverse isochron intercept, inverse isochron age, MDF and J values used.






## Table 3. Major element data (XRF), LOI corrected (in wt%)

| Sample | $SiO_2$ | $TiO_2$ | $Al_2O_3$ | FeO* | MnO | MgO | CaO | $Na_2O$ | $K_2O$ | $TiO_2$ | $P_2O_5$ | LOI |
|---|---|---|---|---|---|---|---|---|---|---|---|---|
| P01A | 64.09 | 0.49 | 16.66 | 3.78 | 0.07 | 3.18 | 5.04 | 3.87 | 2.53 | 0.17 | 0.10 | 0.08 |
| P01B | 77.30 | 0.13 | 12.85 | 0.46 | 0.05 | 0.07 | 0.65 | 4.20 | 4.18 | 0.01 | 0.11 | 0.05 |
| P02 | 51.56 | 1.74 | 18.01 | 8.16 | 0.16 | 4.90 | 8.33 | 4.49 | 2.03 | 0.53 | 0.09 | 0.11 |
| P03 | 75.16 | 0.18 | 13.57 | 1.15 | 0.06 | 0.38 | 1.34 | 4.11 | 3.93 | 0.04 | 0.09 | 0.11 |
| P04 | 67.77 | 0.39 | 16.63 | 2.02 | 0.05 | 0.25 | 1.05 | 4.96 | 6.75 | 0.10 | 0.02 | 0.07 |
| P05 | 74.72 | 0.17 | 13.74 | 1.05 | 0.04 | 0.14 | 0.66 | 4.24 | 5.22 | 0.01 | 0.01 | 0.43 |
| P06 | 80.49 | 0.14 | 10.52 | 0.81 | 0.04 | 0.12 | 0.57 | 3.25 | 4.03 | 0.02 | 0.01 | 0.11 |
| P7 | 64.99 | 0.61 | 16.56 | 3.28 | 0.09 | 1.19 | 3.20 | 3.78 | 5.99 | 0.23 | 0.08 | 0.16 |
| P8 | 74.17 | 0.20 | 13.20 | 1.14 | 0.05 | 0.09 | 0.09 | 1.48 | 9.55 | 0.01 | 0.01 | 0.13 |
| P9 | 78.55 | 0.19 | 11.23 | 1.05 | 0.05 | 0.06 | 0.11 | 2.05 | 6.70 | 0.01 | 0.01 | 0.16 |
| P10A | 60.13 | 0.42 | 20.89 | 2.27 | 0.21 | 0.33 | 1.60 | 7.58 | 6.53 | 0.03 | 0.00 | 1.17 |
| P10B | 60.23 | 0.41 | 20.96 | 2.26 | 0.21 | 0.31 | 1.70 | 7.57 | 6.32 | 0.03 | 0.00 | 0.75 |
| P11 | 65.34 | 0.70 | 17.66 | 3.86 | 0.05 | 1.54 | 0.56 | 1.80 | 8.31 | 0.07 | 0.11 | 0.30 |
| P12 | 59.89 | 0.40 | 21.00 | 2.26 | 0.21 | 0.21 | 1.52 | 7.94 | 6.55 | 0.02 | 0.00 | 0.36 |
| P13 | 65.53 | 0.71 | 17.07 | 3.76 | 0.04 | 0.73 | 2.27 | 3.23 | 6.25 | 0.25 | 0.15 | 0.10 |
| P15 | 76.91 | 0.19 | 13.17 | 0.80 | 0.03 | 0.40 | 0.08 | 0.11 | 8.21 | 0.01 | 0.07 | 0.20 |
| P16 | 68.49 | 0.29 | 16.37 | 1.53 | 0.03 | 0.03 | 0.02 | 1.41 | 11.80 | 0.02 | 0.00 | 0.10 |
| P17 | 53.05 | 1.41 | 18.14 | 6.15 | 0.12 | 5.12 | 5.92 | 3.84 | 5.60 | 0.52 | 0.11 | 0.91 |
| P19 | 65.62 | 0.68 | 16.11 | 3.19 | 0.04 | 0.69 | 0.79 | 1.63 | 10.83 | 0.28 | 0.13 | 0.15 |
| P20 | 69.14 | 0.67 | 16.60 | 0.78 | 0.00 | 0.35 | 0.19 | 1.69 | 10.31 | 0.15 | 0.11 | 0.36 |

**Table 3.** MAJOR ELEMENT DATA (XRF), LOI CORRECTED. (WT%) *No caption.*
