# Peer review of "Volcanism straddling the Mio-Pliocene boundary on Patmos and Chilomodi Islands (SE Aegean Sea): Insights from new $^{40}{\rm Ar}/^{39}{\rm Ar}$ ages"

_Geochronology, 2023_

## Referee Comment (RC2)

**General comment**
This is an interesting and well written manuscript reporting a significant set of Ar/Ar data performed by a new generation multi-collector noble gas mass spectrometer on volcanic rocks from Patmos and the nearby Chilomodi Island (Aegean Sea). Results significantly strengthen the limited set of existing data and permit a detailed reconstruction of volcanism at Patmos, also allowing a revised tectonic reconstruction. Furthermore, data allow general considerations on the reliability of Ar/Ar data from biotite of volcanic rocks. Although for volcanic rock samples it would be preferable to analyse alkali-feldspars, in some chronostratigraphically relevant tephra, biotite is the only K-rich mineral that can be analyzed. Therefore, improving our knowledge about the reliability of biotite in geochronological studies of volcanic rocks, through the comparison of Ar/Ar data from coexisting biotite and sanidine, is of great and broad utility.

The main objection I have on the MS in its present form concerns the uncertainties on Ar/Ar ages (weighted means). The Author state they have conducted an investigation at a high resolution but report uncertainties on the weighted mean ages up to ~2%, quite high for modern geochronology based on multicollector mass spectrometry and using astronomically calibrated reference minerals. First, I recommend that Authors use internal uncertainties (i.e., including analytical errors and uncertainties on the fluence monitor) for inter-sample comparisons of Ar/Ar ages and not the full external uncertainty, which in fact also includes systematic errors (i.e., the uncertainty on the age of the reference material and on $^{40}$K total decay constant) and which affect the calculated ages from the different samples in the same way. If the authors deem it necessary to report the total uncertainty for each age value, I suggest adding it in parentheses after the internal uncertainty. I also suggest to report the first two significant decimal digits both in the age value and in the relative uncertainty. Second, full external uncertainties of up to ~2 % starting from internal uncertainties in most cases of ~0.4-0.5% and using the astronomical calibration for the reference material Fish Canyon sanidine (Kuiper et al. 2008) seem quite high.

My comments mainly focused on Ar/Ar geochronology and I recommend a specific review for the remaining topics. In conclusion, the manuscript needs some work addressing the general comment above and some specific points listed below and after revision, should be publishable in the journal.

**Specific points**
- Line 19 pag. 2: here and through the whole MS replace full external uncertainties with (or add) the 2σ internal uncertainty. The use of the full external uncertainty makes sense only for comparison of Ar ages with those obtained from other radioisotopic systems or non–radioisotopic techniques. See also general comment above.
- Line 132 pag. 6: replace "between 5.7 and 6.0" with "between ~5.7 and ~6.0".
- Lines 133-134: replace "from 5.2 Ma to 5.4 Ma" with "from ~5.4 Ma to ~5.2 Ma".
- Line 135 pag. 6: "rather low K contents"… it is just another phase, anorthoclase.
- Line 149 pag. 6: Why was biotite analysed if it is so altered?
- Line 162 pag. 7: "a biotite age that is ~0.21Ma older" replace with "a biotite age that is 0.219±0.056 Ma older".
- Lines 164-167 pag. 7: "Recoil from K-depleted cleavage zone"… I don't really agree that recoil processes from chloritized levels alone can justify a significant increase in the age of biotite with respect to that of coexisting sanidine. It is well known that chloritization processes induce rejuvenation of total gas ages from biotite (e.g., Roberts et al. 2001; Di Vincenzo et al. 2003).
- Lines 171-172 pag. 7: "on cooling rates at the time of the eruption". In my opinion the cooling rate for an effusive/explosive volcanic rocks is irrelevant in this case. In my opinion the most important cause for older biotite ages with respect to those of coexisting sanidine has not been mentioned: i.e., the presence of excess Ar (parentless $^{40}$Ar), preferentially partitioned into biotite with respect to coexisting sanidine by virtue of its higher mineral/melt partition coefficients (Kelley 2002). I am not saying that extraneous Ar present in biotite P7 cannot be, at least in part, inherited Ar but that excess Ar may be a possible cause and needs to be mentioned.
- line 192 pag. 7: "resp.", make explicit.
- line 193 pag. 8: "we conclude that", this not a conclusion but a finding.

- line 202 pag. 8. The period of activity is 1.370±0.063 Ma (excluding data KB11) or ~1.4 Ma.
- lines 202-203 pag. 8: "with at least three distinct interval of activities", what I see from Ar/Ar data on sanidine (and using ±2σ internal uncertainties) is an onset of volcanism at ~6.5 Ma followed by several pulses down to ~5.2 Ma (see the cumulative probability plot below).

[Figure]

- Line 299 pag. 11: rephrase in "We provide 12 new sanidine and 3 new biotite".
- Fig. 3: Caption, rephrase in "Individual fusion ages with 2σ analytical uncertainties…". I suggest to include in the figure the weighted mean ages followed by the respective internal and full external uncertainties for each mineral separate. Also explain in the caption what are the white dots and why they were excluded (I expect) from the weighted mean calculation.

**References**

Di Vincenzo, G., Viti, C., Rocchi, R. (2003). The effect of chlorite interlayering on $^{40}$Ar–$^{39}$Ar biotite dating: An $^{40}$Ar–$^{39}$Ar laserprobe and TEM investigation of variably chloritised biotites. Contrib. Mineral. Petrol., 145, 643–658.

Kelley S.P. (2002). Excess Ar in K-Ar and Ar–Ar geochronology. Chem. Geol. 188, 1–22.

Kuiper, K.F., Deino, A., Hilgen, F.J., Krijgsman, W., Renne, P.R., Wijbrans, J.R., 2008. Synchronizing rock clocks of Earth history. Science 320, 500–504.

Roberts H.J., Kelley S.P., Dahl P.S. (2001). Obtaining geologically meaningful $^{40}$Ar–$^{39}$Ar ages from altered biotite. Chem. Geol. 172, 277–290.

---

## Author Comment (AC1)

**Response to editor:**

**Dear Klaus Mezger,**

we appreciate the efforts of the two referees and we definitely intend to accommodate the comments where we feel that they strengthen the manuscript. In some cases we disagree with the referees, but in those cases we have provided a motivation as to why we disagree with the referees.

**Please see below, how we plan to address the comments of the two reviewers.**

**Reviewer 1:** *The manuscript by Boehm et al. describes and analyses volcanic rocks from the Island of Patmos in the eastern Aegean Sea region, provides Ar/Ar ages for the volcanic rocks and then attempts to interpret the data in the context of Aegean tectonics. I am not an expert on the geochemistry of volcanic rocks and Ar dating. I assume another reviewer will have the expertise to evaluate those aspects of the manuscript…*
*"and then attempts to interpret the data in the context of Aegean tectonics."*
**Authors:** The Aegean domain and the adjacent west of Anatolia have experienced complex tectonic histories through much of the Cenozoic. For the Aegean tectonic interpretations have been proposed by a limited number of groups, Ring and co-workers being one of these. Alternative tectonic interpretations have been proposed by French teams, with Jolivet the most prominent, Dutch teams led most recently by Van Hinsbergen and Israeli workers around Avigad. In the context of this debate it is emphatically not our intention with our paper to come with a new interpretation of the regional tectonics contributing to the debate on the tectonic evolution of western Anatolia and the Aegean. In our manuscript we intend to provide a simplified snapshot of the area around the Mio-Pliocene boundary, including a broad-brush summary of the tectonics. Here we tried to stay as close as possible to established fact, leaving interpretation to the experts. In the discussion below it is our aim to use valuable comments of the referee to improve our manuscript and at the same time to avoid entering a tectonic discussion that might satisfy the concerns of prof. Ring as a referee, but then possibly triggering debate from his colleagues, professors Van Hinsbergen, Jolivet or Avigad.

*R1: My main concern is how the authors describe and interpret the tectonics of the wider eastern Mediterranean region. I truly believe that they need to seriously improve on that for being able to put their, presumably good and detailed, geochemical and geochronologic results into a tectonic context. Therefore, I am suggesting major rewriting and hence major revisions.*
A: Professor Ring makes a couple of constructive suggestions for improvement in the section below. We will discuss these below.

*R1: In my opinion, the authors need to add a section on 'Regional Setting', or something like that, and need to better understand the regional structure and tectonic development of the region. They would then need to decide, which tectonic processes are important for discussing their data.*
A: Agree. We appreciate the suggestion of professor Ring to add a brief section regional setting. While some of the information is already in the introduction section, the addition of such a section explaining the main features of figure 1 in some more detail would indeed be a valuable addition to the paper. Figure 1 would serve as the basis from which we propose to set up this section

*R1: I would think that the various tectonic units they haphazardly introduced in section 4.5. in the Discussion are not needed.*
A: Agree. We can take 4.5 out and add "Regional Setting" see above.

*R1: They would also need to be clear about the 'asthenospheric window' that underlies western Turkey and how it may, or may not, relate to the Patmos volcanic rocks (as Patmos is about 150 km west of this anomaly).*
A: We will clarify more which literature suggestions we follow for the implications of the seismic anomaly at depth and for the geometry of the gap.

*R1: Western Turkey underwent a distinctly different tectonic evolution then the adjacent Aegean Sea region (see review in for instance Gessner et al. 2013, Gond.Res., https://doi.org/10.1016/j.gr.2013.01.005). One could argue that those different evolutions need some sort of transition zone in between them and Patmos might be part of this transition zone?*
A: We agree with the reviewer that Patmos is situated in a "transition zone". Yes, the Aegean and Western Anatolia have different tectonic evolutions and we follow the suggestion that this difference is compensated by a transfer fault zone (İzmir-Balıkesir Transfer Zone of Uzel et al. or Western Anatolian Transfer Zone of Gessner et al.). We will add information about the transfer zone in the Introduction and discuss tectonic consequences for the local tectonics of Patmos in the Discussion.

*R1: Finally, I consider it also important to better review and define critical age data of the tectonic processes that may help to interpret their data. Usually, literature is cited that does not report any geochronologic data.*

A: Isotopic ages for the basement rocks for both the Cyclades and the Menderes refer to processes much earlier in the history of the region, and are not so relevant to understand the background of the region around the Mio/Pliocene boundary. More useful and pertinent are the paleomagnetic reconstructions, also dated but using the Geomagnetic polarity time scale rather than isotopic dating techniques. The papers of Laj and Kissel and the paper of Duermeijer are pertinent in this respect as dealing with tectonic processes in the pertinent time slice.

*Specific comments:*

*R1: l.20: please be clear what 'rotational crustal forces' are.*

A: We will clarify that we refer to the work of Laj and Kissel and of Duermeijer here, who propose widening of the southern Aegean basin around this time inferring two rotational poles anti clock-wise in the east and clockwise in the west accommodating the extension in the south of the central Aegean.

*R1: l.24: I would mention slab tearing (apparently later in the manuscript referred to as slab fracturing, which is a term not really be used) in the second sentence, which also needs references (Biryol et al. for instance).*

A: We will clarify.

*R1: The next sentence is too simplifying as not only continental platform sediments were subducted. There is also Carboniferous basement, Triassic granites, presumably late Cretaceous oceanic lithosphere etc. subducted to high-P conditions at various times between about 55 and 30 Ma (e.g., Glodny and Ring 2022, ESR, 10.1016/j.earscirev.2021.103883).*

A: We are happy to expand the summary a little further to include the Variscan continental basement rocks and Mesozoic intrusives as well as Mesozoic oceanic crust. We will probably cite a couple of relevant primary references rather than a review article though.

*R1: l.36: Please note that the central Aegean Sea is made up of numerous core complexes, it is NOT one single core complex. Western Turkey is slightly different but the Menderes Massif is also NOT a giant core complex (e.g., review in Gessner et al., 2013).*

A: This is a matter of interpretation, or definition if you want. Indeed since the ground breaking paper of Lister et al. 1984, who first applied the core complex model to the Cyclades, with a special focus on the mantled gneiss dome of Naxos, similar smaller domes have been identified in the region. The island of Ios for example comes to mind but definitely in more Cycladic islands domes features can be found. Having said that, on a much larger scale in the northeast Cyclades the mostly observed sense of shear is top to the northeast, whereas in the western Cyclades the dominant sense of shear is top to the southwest, suggesting movement of an upper plate away from the culmination of the central Cyclades. These shear indicators were found by several groups for the blueschist and greenschist conditions i.e. the earlier phases of exhumation. Subsequently in the central Cyclades stretching lineation in a N-S direction were associated with the later Miocene roll back of the subducting slab placing the whole area in a predominantly extensional stress field. Similarly in the Menderes, low angle normal faulting occurred orogen-wide in the Miocene.

Our point is that both in the Cyclades and in the Menderes around the Mio-Pliocene boundary the time slice of interest to our work, the mode of extension in the two main basement complexes to the east and to the west switched from low-angle normal faulting to horst graben block tectonics.

*R1: l.38. I wonder where 'middle Miocene age' for extreme thinning is coming from? The references provided are not adequate as not a single of these studies reports age data. There are numerous fission-track cooling ages in the central Aegean Sea region (e.g., summaries in Ring et al. 2010 (DOI: 10.1146/annurev.earth.050708.170910), 2017, op cit.); the onset of extension in the Aegean and western Turkey dates back to about 23-34 Ma (see for instance review in Gessner et al., 2013, and references therein).How do you know that Patmos had thinned lithosphere (note that lithosphere involves the crust) by the end of the Miocene?*

A: We should clarify this further. We follow in essence the interpretation of Wijbrans and McDougall 1988, who explained the tectonic history of the central Cyclades in a two stage exhumation process: the first (extension driven) exhumation immediately after the HP metamorphism, bringing the Cycladic rocks back to the middle to upper crust, and the second exhumation, extension in the middle to late Miocene starting around 17 Ma and continuing on well into the Pliocene causing for example the Naxos core complex. Subsequent interpretations as quoted by Ring seem consistent with this interpretation.

*R1: l.40ff. The Cretan Sea basin formed earlier in the middle Miocene, see Drooger, C.W., Meulenkamp, J.E. (1973). Stratigraphic contributions to geodynamics in the Mediterranean area: Crete as a case history. Bulletin of the Geological Society of Greece, 10, pp. 193-200.*

A: While not disputing this fact, we feel that extreme thinning of the Sea of Crete and southern Cyclades occurred in the latest Miocene and Pliocene, based on the paleomagnetic arguments of Laj and Kissel and of Duermeijer.

*R1: l.42ff: The central Menderes metamorphic core complex formed in the Pliocene (e.g., Gessner et al. 2001 (Geology 29 (7), 611-614).*

A: We will further discuss the time line. We feel that the data set of Gessner et al. 2001 leaves ample room for an interpretation that core complex formation started prior to the Pliocene, possibly as early as the middle Miocene following the interpretation of Lips et al. 2001, as AFT data are more commonly interpreted not as dating an event, but rather constraining a process, such as cooling.

*R1: Section 4.2.: The authors take it a bit too far here. Judging from their Fig.3, ages from Patmos are, within error, up to about 7.1 Ma. Saying in l.205 'no evidence for >6.0 Ma' is therefore simply wrong. To me, the ages of Boehm et al. are in agreement with the earlier work.*
A: Our statement is correct: our weighted mean ages do not overlap at the 2-sigma level with their oldest ages of 7.03 ± 0.025 Ma and 7.20 ± 0.025 Ma.

*R1:Section 4.3. is also a bit arm-waving. Apparently, the phonolites are considered important. Only because they are silica undersaturated? Or because they usually occur in intracontinental settings (note that Patmos represents an intracontinental setting)? The only age that is being used in this section is the 14.12 Ma of Altunkaynak et al. 2010 from Foca, an island about 150 km NNE of Patmos. Maybe this age should be reported in the Intro and section 4.3. dropped?*
A:We can expand the argumentation as to why the phonolites are important.

*R1:Section 4.4.I am a bit confused. In section 4.2., it is mentioned that Fytikas et al. (1976) reported ages of about 7 Ma for trachytes. In section 4.4., the authors state that phonolites are the oldest volcanic rocks, followed by trachytes, but the trachytes apparently provide the oldest ages.*
A: Agree, but we do not have the hard data in our dataset to make this point. As these results were obtained using techniques with known difficulties, which we discuss, we have decided not to include these data in our discussions.

*R1: Referring to the very young, intra-plate Kula volcanics is a bit haphazard here. The authors are mixing, also in previous sections of the Discussion, volcanic rocks that developed above the slab tear (asthenospheric window) mapped by Biryol et al. 2011 and the Patmos volcanics, which formed about 150 km W of the slab tear.*
A: This is more a geochemical discussion; however we can add some more explanation why Patmos should be part of a "transiton zone" which is also influenced by the slab gap.

*R1: Section 4.5.The authors should be a bit more careful with their terminology. 'What is 'slab fracturing'? I assume the authors refer to the ca. 300 km wide 'asthenospheric window' in western Turkey, which a slow wave speed anomaly that is commonly interpreted as a tear in the African plate (Biryol et al. 2011).* A: Agree

*R1: In l.248-254 a tectonic subdivision of the Aegean/Menderes region is casually weaved in and terms like Lower and Upper Cycladic Blueschist Nappe, Amorgos unit, Menderes cover sequence, Ören unit, Afyon unit, Dilek Nappe and Trans Cycladic thrust are being used without any context and explanation. This is absolutely not acceptable and utterly confusing. The Lower (and Upper) Cycladic Blueschist Unit (and the Trans Cycladic Thrust) are a concept, first introduced by Grasemann et al. 2018, GSAB (DOI: 10.1130/B31731.1) and these two units and the thrust are defined in the western Cyclades. Whether or not the Lower and Upper Cycladic Blueschist Units can also be distinguished in the eastern Aegean Sea region is unknown and the speculative correlations by Roche et al. (2019, op. cit.) are not being backed-up by data (see Glodny and Ring 2022 for a different view). Menderes cover sequence is something that has been introduced in the middle of the last century but not being a sound concept these days anymore. The standard reader does not have a clue what Amorgos unit refers to (Laskari et al. 2022, https://doi.org/10.1016/j.gr.2022.02.007). All these units would need to be properly introduced and discussed. In the current manuscript the reader is wholly confused and lost.*
*L.256: Now exhumation is early to middle Miocene in age. In the Intro 'extension' was supposed to be middle Miocene in age. Note that most of the exhumation of the Cycladic blueschists occurred much before the Miocene and had nothing to do with extensional deformation.*
A: Following Wijbrans and McDougall 1988, exhumation is interpreted as two-stage, the first early stage was completed by the beginning of the Oligocene, the second stage, unconnected ot the first, started probably around 17 Ma on Naxos, and is related to the formation of the core complex there. Most subsequent literature follows this line of interpretation.

*R1: In the following (l.256-264), things become more than vague. The phase of relatively young 'N-S shortening' does most probably not exist and is not being accepted by people working in the Menderes massif. Why is the North Anatolian fault and the Peleponnese important here? There is no tectonic context provided and all things of things are thrown into the mix.*
*L.260: see Gessner et al. 2001 (Geology, see above) for actual age data of this renewed phase of N-S extension, which created the Central Menderes Metamorphic core complex. Neither Westaway nor Bozkurt and Mittwede report any age data.*
*L.267. The E-W shortening phase in Samos is actually well dated between <8.6 and c. 9 Ma by Ring et al. (1999, op cit.).*
*l.266ff: Most of the info here is literature review and should go into an introductory section on 'Regional Setting'. As mentioned above, I believe the authors need to completely rewrite section 4.5. if they wish to provide a tectonic interpretation of the Patmos volcanics (see general comments above).*

A: Agree. As mentioned above we are willing to delete 4.5 and write Regional Setting: The main purpose of this section will be to review dating of tectonic processes in the local area of Patmos, while the focus of the paper is the dating of magmatic rocks and not the discussion of tectonic evolutions.

R1:Fig.1: The map is largely ok; however, there are a few structures and tectonic subdivisions that would need to be discussed properly as they are controversial. The Mid-Cycladic lineament (MCL in Fig.1) is a concept first proposed (if I remember correctly) by Walcott and White 1988, https://doi.org/10.1016/S0040-1951(98)00182-6. Nobody has ever seen this lineament and there is no direct evidence for its existence. It is just a concept a few people believe in and use, while most people simply do not believe in its existence. This needs to be made clearer. One cannot create a fact out of something that is not understood and controversial. The IBTZ (not explained in the figure caption) is a similar problem. The same is true for Lower (and Upper) Cycladic Blueschist Unit (see above).
A: We will modify figure and figure caption. We will not enter a broad tectonic discussion, but rather use references for the goal of defining the local tectonic setting of Patmos.

**Reviewer 2**
Reviewer2: The main objection I have on the MS in its present form concerns the uncertainties on Ar/Ar ages (weighted means). The Author state they have conducted an investigation at a high resolution but report uncertainties on the weighted mean ages up to ~2%, quite high for modern geochronology based on multicollector mass spectrometry and using astronomically calibrated reference minerals. First, I recommend that Authors use internal uncertainties (i.e., including analytical errors and uncertainties on the fluence monitor) for inter-samplecomparisons of Ar/Ar ages and not the full external uncertainty, which in fact also includes systematic errors (i.e., the uncertainty on the age of the reference material and on 40K total decay constant) and which affect the calculated ages from the different samples in the same way. If the authors deem it necessary to report the total uncertainty for each age value, I suggest adding it in parentheses after the internal uncertainty. I also suggest to report the first two significant decimal digits both in the age value and in the relative uncertainty.
Authors: We will report all our ages with the analytical uncertainty, including uncertainty in J and add full external error between parentheses. Note, that in table 2 we already report both. We will add the 1 extra significant decimal. In figure 3 we show 2σ analytical uncertainties for individual analyses. We will add this information to the figure caption.

R2: Second, full external uncertainties of up to ~2 % starting from internal uncertainties in most cases of ~0.4-0.5% and using the astronomical calibration for the reference material Fish Canyon sanidine (Kuiper et al. 2008) seem quite high.
A: This is correct when using Min et al, 2001 decay constants in combination with 28.201Ma age for FCs. Sometimes the FCs 28.201Ma age is combined with Steiger and Jager (1997), this is incorrect, but leads to smaller external uncertainties.

Specific points
R2: Line 19 pag. 2: here and through the whole MS replace full external uncertainties with (or add) the 2σ internal uncertainty. The use of the full external uncertainty makes sense only for comparison of Ar ages with those obtained from other radioisotopic systems or non–radioisotopic techniques. See also general comment above.
A: Agree, will be adapted

R2: Line 132 pag. 6: replace “between 5.7 and 6.0” with “between ~5.7 and ~6.0”.
A:Agree, will be adapted

R2: Lines 133-134: replace “from 5.2 Ma to 5.4 Ma” with “from ~5.4 Ma to ~5.2 Ma”.
A: Agree, will be adapted

R2: Line 135 pag. 6: “rather low K contents”… it is just another phase, anorthoclase.
A: Will be rephrased

R2: Line 149 pag. 6: Why was biotite analysed if it is so altered?
A: Will be rephrased: based on visual inspection we did not conclude that biotites were altered, but concluded that based on the Ar/Ar results.

R2: Line 162 pag. 7: “a biotite age that is ~0.21Ma older” replace with “a biotite age that is 0.219±0.056 Ma older”.
A: Agree, will be adapted

*R2: Lines 164-167 pag. 7: "Recoil from K-depleted cleavage zone"… I don't really agree that recoil processes from chloritized levels alone can justify a significant increase in the age of biotite with respect to that of coexisting sanidine. It is well known that chloritization processes induce rejuvenation of total gas ages from biotite (e.g., Roberts et al. 2001; Di Vincenzo et al. 2003).*

A: While alteration processes in many cases will tend to lower the total fusion age of a sample, there are exceptions. The case of biotite alteration into K-free alteration products such as chlorite or clay minerals, forming along the a-b planes in the biotite crystal structure is one of these. This is the mechanism that we are calling upon here. In this case the 39Ar formed by neutron irradiation recoils into the alteration zones within the original biotite crystals, and while they become lost for the biotite apparent ages increase. The process has been described in the literature, in addition to Roberts and Di Vinzenzo as cited by the referee, it was first systematically described by Hess et al in 1987. All three studies note that chloritization can produce systematically too old ages due to excess 39Ar recoil into the K-poor chlorite lamellae. The examples of Roberts and Di Vincenzo show in a ddition to this process substantial losses of radiogenic argon. We note that the materials used by Di Vincenzo and Roberts for their experiments may not be representative for the case of very mild alteration as one might expect for volcanic biotites. Shereas the materials used by Hess et al. might be more representative. We have seen ourselves systematically old biotite ages when compared with co-genetic sanidines fairly consistently during ca 25 years of carrying out tephrachronology studies (e.g. Steenbrink et al. 1999).

J.C. Hess, H.J. Lippolt, R. Wirth (1987) Interpretation of 40Ar/39Ar spectra of biotites: Evidence from hydrothermal degassing experiments and TEM studies Chemical Geology 66, 137-149.

*R2: Lines 171-172 pag. 7: "on cooling rates at the time of the eruption". In my opinion the cooling rate for an effusive/explosive volcanic rocks is irrelevant in this case. In my opinion the most important cause for older biotite ages with respect to those of coexisting sanidine has not been mentioned: i.e., the presence of excess Ar (parentless 40Ar), preferentially partitioned into biotite with respect to coexisting sanidine by virtue of itshigher mineral/melt partition coefficients (Kelley 2002). I am not saying that extraneous Ar present in biotite P7 cannot be, at least in part, inherited Ar but that excess Ar may be a possible cause and needs to be mentioned.*

A: Will be rephrased. We hint at that in this section, but did not write this down sufficiently clear. We like to note that all isochron intercepts (based on the fusion experiments) do not deviate from atmospheric argon and excess argon might be unlikely in these samples.

*R2: line 192 pag. 7: "resp.", make explicit.*

A: Agree, will be adapted

*R2: line 193 pag. 8: "we conclude that", this not a conclusion but a finding.*

A: Agree, will be adapted

*R2: line 202 pag. 8. The period of activity is 1.370±0.063 Ma (excluding data KB11) or ~1.4 Ma.*

A: Will be rephrased: The different between the oldest and youngest reported age is 1.370±0.063 Ma (excluding data KB11) or ~1.4 Ma.

*R2: lines 202-203 pag. 8: "with at least three distinct interval of activities", what I see from Ar/Ar data on sanidine (and using ±2σ internal uncertainties) is an onset of volcanism at ~6.5 Ma followed by several pulses down to ~5.2 Ma (see the cumulative probability plot below).*

A: Will be rephrased: we did not provide a cumulative density plot, because this suggests an accuracy of individual dates that is not realistic. Based on visual inspection of figure 3 including all data one could argue there is an active phase from ~6.7Ma - ~6.4Ma (Patmos), from ~6.1Ma - ~5.5Ma (Patmos) and from ~5.5 Ma – ~5.1 Ma (Chilomodi). These are the "at least three distinct phases" we mention in the manuscript. Discarding all biotite data because of recoil or alteration issues is a valid argument. When plotting the remaining data in a density plot 7 peaks appear (see review). Combining the density plot with our figure 3 shows that sample KB12 contributes to the peak around 5.6Ma and KB13 around 5.7Ma. This suggest two different phases, but the dated minerals are from the same sample. Based on the sample complexity we prefer not to make the claim that these are two pulses, but rather prefer to state there is at least one pulse.

---

## Author Response (AR1)

We appreciate the efforts of the two referees and we accommodated most of the comments where we felt that they strengthen the manuscript. In some cases we disagree with the referees, but in those cases we have provided a motivation as to why we disagree with the referees.

**Please see below, how we address the comments.**

**Reviewer 1:** *The manuscript by Boehm et al. describes and analyses volcanic rocks from the Island of Patmos in the eastern Aegean Sea region, provides Ar/Ar ages for the volcanic rocks and then attempts to interpret the data in the context of Aegean tectonics. I am not an expert on the geochemistry of volcanic rocks and Ar dating. I assume another reviewer will have the expertise to evaluate those aspects of the manuscript...*
*"and then attempts to interpret the data in the context of Aegean tectonics."*
**Authors:** The Aegean domain and the adjacent west of Anatolia have experienced complex tectonic histories through much of the Cenozoic. For the Aegean tectonic interpretations have been proposed by a limited number of groups. In the context of this debate it is not our intention to propose a new interpretation of the regional tectonics and add another view to the tectonic evolution of western Anatolia and the Aegean. In our manuscript we intend to provide a simplified snapshot of the area around the Mio-Pliocene boundary, including a broad-brush summary of the tectonics. Here we tried to stay as close as possible to empirical data, leaving interpretation to the experts. In the discussion below it is our aim to use valuable comments of the referee to improve our manuscript and at the same time to avoid entering a tectonic discussion between different groups working on the regional tectonics.

*R1: My main concern is how the authors describe and interpret the tectonics of the wider eastern Mediterranean region. I truly believe that they need to seriously improve on that for being able to put their, presumably good and detailed, geochemical and geochronologic results into a tectonic context. Therefore, I am suggesting major rewriting and hence major revisions.*
A: Thanks for constructive suggestions for improvement in the section below. We will discuss these below.

*R1: In my opinion, the authors need to add a section on 'Regional Setting', or something like that, and need to better understand the regional structure and tectonic development of the region. They would then need to decide, which tectonic processes are important for discussing their data.*
A: Agree. We appreciate the suggestion to add a brief section "Regional Setting". While some of the information is already in the introduction section, the addition of such a section explaining the main features of figure 1 in some more detail is indeed a valuable addition to the paper.

*R1: I would think that the various tectonic units they haphazardly introduced in section 4.5. in the Discussion are not needed.*
A: Agree. We adjusted section 4.5 and added this information in the section "Regional Setting" (see above).

*R1: They would also need to be clear about the 'asthenospheric window' that underlies western Turkey and how it may, or may not, relate to the Patmos volcanic rocks (as Patmos is about 150 km west of this anomaly).*
A: We now better explain the literature we used for the implications of the seismic anomaly and for the geometry of the gap, which suggest that the Eastern Aegean including Patmos are above the (shallow parts of the) anomaly (see Govers and Fichnter 2016).
See line 37 "The presence of an approximately vertical gap between neighbouring Hellenic and Cyprus slabs is seen in tomographic and seismic data. The anomaly of the gap underlies Western Anatolia at depth (e.g. Biryol et al., 2011) and reaches into the Eastern Aegean in shallower levels (< 25km, Govers and Fichtner, 2016). Because of these indications of geometry the gap possibly provides a channel for deeper asthenospheric magma sources also in the eastern Aegean. In addition to the gap between the two slabs, there is speculation about the tearing of the Hellenic slab, which has been called upon for the interpretation of geochemical signals of volcanics in the eastern Aegean (Ersoy and Palmer, 2013; Klaver et al., 2016a; Palmer et al., 2019; Uzel et al., 2020) "

*R1: Western Turkey underwent a distinctly different tectonic evolution then the adjacent Aegean Sea region (see review in for instance Gessner et al. 2013, Gond.Res., https://doi.org/10.1016/j.gr.2013.01.005). One could argue that those different evolutions need some sort of transition zone in between them and Patmos might be part of this transition zone?*
A: We agree with the reviewer that Patmos is situated in a "transition zone". Yes, the Aegean and Western Anatolia have different tectonic evolutions and we follow the suggestion that this difference is compensated by a transfer fault zone (İzmir-Balıkesir Transfer Zone of Uzel et al. or Western Anatolian Transfer Zone of Gessner et al.). We added information about the transfer zone in the Introduction and discuss tectonic consequences for the local tectonics of Patmos in the Discussion.
See line 63: "In late Miocene to Pliocene, Patmos was situated in a tectonic transition zone, where transfer faulting also compensated the transition between the Aegean and Western Anatolian volcanic provinces."

*R1: Finally, I consider it also important to better review and define critical age data of the tectonic processes that may help to interpret their data. Usually, literature is cited that does not report any geochronologic data.*
A: Isotopic ages for the basement rocks for both the Cyclades and the Menderes refer to processes much earlier in the history of the region, and are not so relevant to understand the background of the region around the Mio/Pliocene boundary. More useful and pertinent are the paleomagnetic reconstructions, and sediments dated using the geomagnetic polarity time scale rather than isotopic dating techniques. The papers of Laj and Kissel and the paper of Duermeijer are pertinent in this respect as they are dealing with tectonic processes in the pertinent time slice.

*Specific comments:*
*R1: l.20: please be clear what 'rotational crustal forces' are.*
A: We clarified that we refer to the work of Laj and Kissel and of Duermeijer here, who propose widening of the southern Aegean basin around this time inferring two rotational poles anti clock-wise in the east and clockwise in the west accommodating the extension in the south of the central Aegean.
Line 21: "and rotational crustal forces that also caused the widening of the south Aegean basin due to two opposite rotational poles in the east and west and due to roll back of the subducting slab south of Crete."

*R1: l.24: I would mention slab tearing (apparently later in the manuscript referred to as slab fracturing, which is a term not really be used) in the second sentence, which also needs references (Biryol et al. for instance).*
A: We favor opening of an gap between two slabs over the scenario of a slab tear and therefore use this term (see above at asthenospheric window)

*R1: The next sentence is too simplifying as not only continental platform sediments were subducted. There is also Carboniferous basement, Triassic granites, presumably late Cretaceous oceanic lithosphere etc. subducted to high-P conditions at various times between about 55 and 30 Ma (e.g., Glodny and Ring 2022, ESR, 10.1016/j.earscirev.2021.103883).*
A: We expanded the summary a little further and included the Variscan continental basement rocks and Mesozoic intrusives as well as Mesozoic oceanic crust. We cited a couple of primary references (Jacobshagen).
Line 27: "The subduction zone magmatism was governed by the composition of the subducting plate (Variscan continental basement rocks, intrusives, continental and oceanic crust and sediment covers; e.g. Jacobshagen et al., 1978, Jacobshagen, 1994), slab derived –fluids and depth of magma generation, including assimilation and fractionation plus magma mixing."

*R1: l.36: Please note that the central Aegean Sea is made up of numerous core complexes, it is NOT one single core complex. Western Turkey is slightly different but the Menderes Massif is also NOT a giant core complex (e.g., review in Gessner et al., 2013).*
A: This is a matter of interpretation, or definition if you want. Indeed since the ground breaking paper of Lister et al. 1984, who first applied the core complex model to the Cyclades, with a special focus on the mantled gneiss dome of Naxos, similar smaller domes have been identified in the region. The island of Ios for example comes to mind but definitely in more Cycladic islands dome features can be found. Having said that, on a much larger scale in the northeast Cyclades the mostly observed sense of shear is top to the northeast, whereas in the western Cyclades the dominant sense of shear is top to the southwest, suggesting movement of an upper plate away from the culmination of the central Cyclades. These shear indicators were found by several groups for the blueschist and greenschist conditions i.e. the earlier phases of exhumation. Subsequently in the central Cyclades stretching lineation in a N-S direction was associated with the later Miocene roll-back of the subducting slab placing the whole area in a predominantly extensional stress field. Similarly in the Menderes, low-angle normal faulting occurred orogen-wide in the Miocene and we are aware existence of the different detachment surfaces between different units through the Menderes Massif. Since it's out of the scope of our study, we follow the nomenclature of literature. Our point is that both in the Cyclades and the Menderes around the Mio-Pliocene boundary the time slice of interest to our work, the mode of extension in the two main basement complexes to the east and to the west switched from low-angle normal faulting to horst graben block tectonics.

*R1: l.38. I wonder where 'middle Miocene age' for extreme thinning is coming from? The references provided are not adequate as not a single of these studies reports age data. There are numerous fission-track cooling ages in the central Aegean Sea region (e.g., summaries in Ring et al. 2010 (DOI: 10.1146/annurev.earth.050708.170910), 2017, op cit.); the onset of extension in the Aegean and western Turkey dates back to about 23-34 Ma (see for instance review in Gessner et al., 2013, and references therein).How do you know that Patmos had thinned lithosphere (note that lithosphere involves the crust) by the end of the Miocene?*
A: We tried to clarify this further. We follow in essence the interpretation of Wijbrans and McDougall 1988, who explained the tectonic history of the central Cyclades in a two-stage exhumation process: the first (extension driven) exhumation immediately after the HP metamorphism, bringing the Cycladic rocks back to the middle to upper crust, and the second exhumation, extension in the middle to late Miocene starting around 17 Ma and continuing well into the Pliocene causing, for example, the Naxos core complex. Subsequent interpretations as quoted seem consistent with this interpretation.

*R1: l.40ff. The Cretan Sea basin formed earlier in the middle Miocene, see Drooger, C.W., Meulenkamp, J.E. (1973). Stratigraphic contributions to geodynamics in the Mediterranean area: Crete as a case history. Bulletin of the Geological Society of Greece, 10, pp. 193-200.*

A: While not disputing this fact, we feel that extreme thinning of the Sea of Crete and southern Cyclades occurred in the latest Miocene and Pliocene, based on the paleomagnetic arguments of Laj and Kissel and of Duermeijer.

*R1: l.42ff: The central Menderes metamorphic core complex formed in the Pliocene (e.g., Gessner et al. 2001 (Geology 29 (7), 611-614).*

A: We feel that the data set of Gessner et al. 2001 leaves ample room for an interpretation that core complex formation started prior to the Pliocene, possibly as early as the middle Miocene following the interpretation of Lips et al. 2001, as AFT data are more commonly interpreted not as dating an event, but rather constraining a process, such as cooling. The Menderes core complex started to form in Oligocene and to uplift since then. Gessner et al (2013) showed that the exhumation of the Menderes core complex started in Early Miocene and occurred at least in two stages; Middle Miocene (cover series) and Late Miocene-Recent (central part). On the other hand, the first metamorphic clasts in the supra-detachment basins were found in the Lower Miocene strata (Sözbilir, 2005). This shows that (i) the exhumation of the core complexes is not linear but spatiotemporal, or (ii) in literature a lot of different scenarios are presented. In any approach, there are slight differences between the Cyclades and Menderes in terms of metamorphism and exhumation history, therefore the İBTZ (or WASZ) must have played an important role during their subduction and exhumation processes. We clarified this difference in the text.

*R1: Section 4.2.: The authors take it a bit too far here. Judging from their Fig.3, ages from Patmos are, within error, up to about 7.1 Ma. Saying in l.205 'no evidence for >6.0 Ma' is therefore simply wrong. To me, the ages of Boehm et al. are in agreement with the earlier work.*

A: Our statement is correct: our weighted mean ages do not overlap at the 2-sigma level with their oldest ages of 7.03 ± 0.025 Ma and 7.20 ± 0.025 Ma.

*R1:Section 4.3. is also a bit arm-waving. Apparently, the phonolites are considered important. Only because they are silica undersaturated? Or because they usually occur in intracontinental settings (note that Patmos represents an intracontinental setting)? The only age that is being used in this section is the 14.12 Ma of Altunkaynak et al. 2010 from Foca, an island about 150 km NNE of Patmos. Maybe this age should be reported in the Intro and section 4.3. dropped?*
A:We included this part to the Introduction.

*R1:Section 4.4.I am a bit confused. In section 4.2., it is mentioned that Fytikas et al. (1976) reported ages of about 7 Ma for trachytes. In section 4.4., the authors state that phonolites are the oldest volcanic rocks, followed by trachytes, but the trachytes apparently provide the oldest ages.*

A: Agree, but we do not have the hard data in our dataset to make this point. As these results were obtained using techniques with known difficulties, which we discuss, we have decided not to include these data in our discussions.

*R1: Referring to the very young, intra-plate Kula volcanics is a bit haphazard here. The authors are mixing, also in previous sections of the Discussion, volcanic rocks that developed above the slab tear (asthenospheric window) mapped by Biryol et al. 2011 and the Patmos volcanics, which formed about 150 km W of the slab tear.*

A: This is more a geochemical discussion; however we added some short explanation why Patmos should be part of a "transiton zone" which is also influenced by the slab gap.

*R1: Section 4.5.The authors should be a bit more careful with their terminology. 'What is 'slab fracturing'? I assume the authors refer to the ca. 300 km wide 'asthenospheric window' in western Turkey, which a slow wave speed anomaly that is commonly interpreted as a tear in the African plate (Biryol et al. 2011).* A: Agree

*R1: In l.248-254 a tectonic subdivision of the Aegean/Menderes region is casually weaved in and terms like Lower and Upper Cycladic Blueschist Nappe, Amorgos unit, Menderes cover sequence, Ören unit, Afyon unit, Dilek Nappe and Trans Cycladic thrust are being used without any context and explanation. This is absolutely not acceptable and utterly confusing. The Lower (and Upper) Cycladic Blueschist Unit (and the Trans Cycladic Thrust) are a concept, first introduced by Grasemann et al. 2018, GSAB (DOI: 10.1130/B31731.1) and these two units and the thrust are defined in the western Cyclades. Whether or not the Lower and Upper Cycladic Blueschist Units can also be distinguished in the eastern Aegean Sea region is unknown and the speculative correlations by Roche et al. (2019, op. cit.) are not being backed-up by data (see Glodny and Ring 2022 for a different view). Menderes cover sequence is something that has been introduced in the middle of the last century but not being a sound concept these days anymore. The standard reader does not have a clue what Amorgos unit refers to (Laskari et al. 2022, https://doi.org/10.1016/j.gr.2022.02.007). All these units would need to be properly introduced and discussed. In the current manuscript the reader is wholly confused and lost.*
*We are focusing on the tectonics features and regimes influencing volcanism and Patmos at the* Mio-Pliocene boundary *and avoiding at the same time to enter the discussion of Cycladic and Menderes units*, leaving interpretation to the experts.

*R1: L.256: Now exhumation is early to middle Miocene in age. In the Intro 'extension' was supposed to be middle Miocene in age. Note that most of the exhumation of the Cycladic blueschists occurred much before the Miocene and had nothing to do with extensional deformation.*

A: Following Wijbrans and McDougall 1988, exhumation is interpreted as two-stage, the first early stage was completed by the beginning of the Oligocene, the second stage, unconnected ot the first, started probably around 17 Ma on Naxos, and is related to the formation of the core complex there.We revised this in our manuscript to make more clear what we mean. Most subsequent literature follows this line of interpretation.

*R1: In the following (l.256-264), things become more than vague. The phase of relatively young 'N-S shortening' does most probably not exist and is not being accepted by people working in the Menderes massif. Why is the North Anatolian fault and the Peleponnese important here? There is no tectonic context provided and all things of things are thrown into the mix. L.260: see Gessner et al. 2001 (Geology, see above) for actual age data of this renewed phase of N-S extension, which created the Central Menderes Metamorphic core complex. Neither Westaway nor Bozkurt and Mittwede report any age data. L.267. The E-W shortening phase in Samos is actually well dated between <8.6 and c. 9 Ma by Ring et al. (1999, op cit.). l.266ff: Most of the info here is literature review and should go into an introductory section on 'Regional Setting'. As mentioned above, I believe the authors need to completely rewrite section 4.5. if they wish to provide a tectonic interpretation of the Patmos volcanics (see general comments above).*

A: Agree. As mentioned above we are restructuring 4.5 and write a new Regional Setting in the introduction: The main purpose of this section is to review dating of tectonic processes in the local area of Patmos, while the focus of the paper is the dating of magmatic rocks and not the discussion of tectonic evolutions. Section is revised.

*R1:Fig.1: The map is largely ok; however, there are a few structures and tectonic subdivisions that would need to be discussed properly as they are controversial. The Mid-Cycladic lineament (MCL in Fig.1) is a concept first proposed (if I remember correctly) by Walcott and White 1988, https://doi.org/10.1016/S0040-1951(98)00182-6. Nobody has ever seen this lineament and there is no direct evidence for its existence. It is just a concept a few people believe in and use, while most people simply do not believe in its existence. This needs to be made clearer. One cannot create a fact out of something that is not understood and controversial. The IBTZ (not explained in the figure caption) is a similar problem. The same is true for Lower (and Upper) Cycladic Blueschist Unit (see above).*

A: We modified figure and figure caption. We acknowledge there are different views about the existence of the MCL, however we follow the publications about MCL (e.g. Lykousis et al., 1995; Pe-Piper et al., 2002; Tirel et al., 2009; Danele et al., 2011; Philippon et al., 2012; Cetinkaplan et al., 2020). We do not want to enter a broad tectonic discussion, but rather use references for the goal of defining the local tectonic setting of Patmos.

**Reviewer 2**

*Reviewer2: The main objection I have on the MS in its present form concerns the uncertainties on Ar/Ar ages (weighted means). The Author state they have conducted an investigation at a high resolution but report uncertainties on the weighted mean ages up to ~2%, quite high for modern geochronology based on multicollector mass spectrometry and using astronomically calibrated reference minerals. First, I recommend that Authors use internal uncertainties (i.e., including analytical errors and uncertainties on the fluence monitor) for inter-samplecomparisons of Ar/Ar ages and not the full external uncertainty, which in fact also includes systematic errors (i.e., the uncertainty on the age of the reference material and on 40K total decay constant) and which affect the calculated ages from the different samples in the same way. If the authors deem it necessary to report the total uncertainty for each age value, I suggest adding it in parentheses after the internal uncertainty. I also suggest to report the first two significant decimal digits both in the age value and in the relative uncertainty.*

A: We report all our ages with the analytical uncertainty, including uncertainty in J and add full external error between parentheses. Note, that in table 2 we already reported both. We added the 1 extra significant decimal. In figure 3 we show 2σ analytical uncertainties for individual analyses. We added this information to the figure caption.

*R2: Second, full external uncertainties of up to ~2 % starting from internal uncertainties in most cases of ~0.4-0.5% and using the astronomical calibration for the reference material Fish Canyon sanidine (Kuiper et al. 2008) seem quite high.*

A: This is correct when using Min et al, 2001 decay constants in combination with 28.201Ma age for FCs. Sometimes the FCs 28.201Ma age is combined with Steiger and Jager (1978), this is incorrect, but leads to smaller external uncertainties.

*Specific points*
*R2: Line 19 pag. 2: here and through the whole MS replace full external uncertainties with (or add) the 2σ internal uncertainty. The use of the full external uncertainty makes sense only for comparison of Ar ages with those obtained from other radioisotopic systems or non–radioisotopic techniques. See also general comment above.*

A: Agree, is adapted

*R2: Line 132 pag. 6: replace "between 5.7 and 6.0" with "between ~5.7 and ~6.0".*
A: Agree, is adapted

*R2: Lines 133-134: replace "from 5.2 Ma to 5.4 Ma" with "from ~5.4 Ma to ~5.2 Ma".*
*A:* Agree, is adapted

*R2: Line 135 pag. 6: "rather low K contents"… it is just another phase, anorthoclase.*
A: Is rephrased
"P3 contains anorthoclase, with K contents ranging between ~2.0-2.6 wt.% K"

*R2: Line 149 pag. 6: Why was biotite analysed if it is so altered?*
A: Is rephrased: based on visual inspection we did not conclude that biotites were altered, but concluded that based on the Ar/Ar results.
"Based on visual inspection the biotites appeared to have sufficient quality, but based on the Ar/Ar ages we conclude the biotites were too altered and did not yield reliable ages."

*R2: Line 162 pag. 7: "a biotite age that is ~0.21Ma older" replace with "a biotite age that is 0.219±0.056 Ma older".*
A: Agree, is adapted

*R2: Lines 164-167 pag. 7: "Recoil from K-depleted cleavage zone"… I don't really agree that recoil processes from chloritized levels alone can justify a significant increase in the age of biotite with respect to that of coexisting sanidine. It is well known that chloritization processes induce rejuvenation of total gas ages from biotite (e.g., Roberts et al. 2001; Di Vincenzo et al. 2003).*
A: While alteration processes in many cases will tend to lower the total fusion age of a sample, there are exceptions. The case of biotite alteration into K-free alteration products such as chlorite or clay minerals, forming along the a-b planes in the biotite crystal structure is one of these. This is the mechanism that we are calling upon here. In this case the 39Ar formed by neutron irradiation recoils into the alteration zones within the original biotite crystals, and while they become lost for the biotite apparent ages increase. The process has been described in the literature. In addition to Roberts and Di Vinzenzo as cited by the referee, it was first systematically described by Hess et al in 1987. All three studies note that chloritization can produce systematically too old ages due to excess 39Ar recoil into the K-poor chlorite lamellae. The examples of Roberts and Di Vincenzo show in addition to this process substantial losses of radiogenic argon. We note that the materials used by Di Vincenzo and Roberts for their experiments may not be representative for the case of very mild alteration as one might expect for volcanic biotites, whereas the materials used by Hess et al. might be more representative. We have seen ourselves systematically old biotite ages when compared with co-genetic sanidines fairly consistently during ca 25 years of carrying out tephrachronology studies (e.g. Steenbrink et al. 1999).
J.C. Hess, H.J. Lippolt, R. Wirth (1987) Interpretation of 40Ar/39Ar spectra of biotites: Evidence from hydrothermal degassing experiments and TEM studies Chemical Geology 66, 137-149.

*R2: Lines 171-172 pag. 7: "on cooling rates at the time of the eruption". In my opinion the cooling rate for an effusive/explosive volcanic rocks is irrelevant in this case. In my opinion the most important cause for older biotite ages with respect to those of coexisting sanidine has not been mentioned: i.e., the presence of excess Ar (parentless 40Ar), preferentially partitioned into biotite with respect to coexisting sanidine by virtue of itshigher mineral/melt partition coefficients (Kelley 2002). I am not saying that extraneous Ar present in biotite P7 cannot be, at least in part, inherited Ar but that excess Ar may be a possible cause and needs to be mentioned.*
A: Is rephrased. We like to note that all isochron intercepts (based on the fusion experiments) do not deviate from atmospheric argon and excess argon might be unlikely in these samples.

*R2: line 192 pag. 7: "resp.", make explicit.*
A: Agree, is adapted

*R2: line 193 pag. 8: "we conclude that", this not a conclusion but a finding.*
A: Agree, is adapted

*R2: line 202 pag. 8. The period of activity is 1.370±0.063 Ma (excluding data KB11) or ~1.4 Ma.*
A: Is rephrased: The different between the oldest and youngest reported age is 1.370±0.063 Ma (excluding data KB11) or ~1.4 Ma.

*R2: lines 202-203 pag. 8: "with at least three distinct interval of activities", what I see from Ar/Ar data on sanidine (and using ±2σ internal uncertainties) is an onset of volcanism at ~6.5 Ma followed by several pulses down to ~5.2 Ma (see the cumulative probability plot below).*

A: Is rephrased: we did not provide a cumulative density plot, because this suggests an accuracy of individual dates that is not realistic. Based on visual inspection of figure 3 including all data one could argue there is an active phase from ~6.7Ma - ~6.4Ma (Patmos), from ~6.1Ma - ~5.5Ma (Patmos) and from ~5.5 Ma – ~5.1 Ma (Chilomodi). These are the "at least three distinct phases" we mention in the manuscript. Discarding all biotite data because of recoil or alteration issues is a valid argument. When plotting the remaining data in a density plot 7 peaks appear (see review). Combining the density plot with our figure 3 shows that sample KB12 contributes to the peak around 5.6Ma and KB13 around 5.7Ma. This suggest two different phases, but the dated minerals are from the same sample. Based on the sample complexity we prefer not to make the claim that these are two pulses, but rather prefer to state there is at least one pulse.

"The accuracy, sample complexity and number of the individual dates does not allow further distinction of intervals, although there might be more."